# Tau exacerbates excitotoxic brain damage in an animal model of stroke

Mian Bi[1], Amadeus Gladbach[1], Janet van Eersel[1], Arne Ittner[1], Magdalena Przybyla[1], Annika van Hummel[1], Sook Wern Chua[1], Julia van der Hoven[1], Wei S. Lee[1], Julius Müller [2,3], Jasneet Parmar[4], Georg von Jonquieres[4], Holly Stefen[5], Ernesto Guccione[2,6], Thomas Fath[5,7], Gary D. Housley [4], Matthias Klugmann [4], Yazi D. Ke[1] & Lars M. Ittner[1,8,9]

Neuronal excitotoxicity induced by aberrant excitation of glutamatergic receptors contributes to brain damage in stroke. Here we show that tau-deficient (tau$^{-/-}$) mice are profoundly protected from excitotoxic brain damage and neurological deficits following experimental stroke, using a middle cerebral artery occlusion with reperfusion model. Mechanistically, we show that this protection is due to site-specific inhibition of glutamate-induced and Ras/ERK-mediated toxicity by accumulation of Ras-inhibiting SynGAP1, which resides in a post-synaptic complex with tau. Accordingly, reducing SynGAP1 levels in tau$^{-/-}$ mice abolished the protection from pharmacologically induced excitotoxicity and middle cerebral artery occlusion-induced brain damage. Conversely, over-expression of SynGAP1 prevented excitotoxic ERK activation in wild-type neurons. Our findings suggest that tau mediates excitotoxic Ras/ERK signaling by controlling post-synaptic compartmentalization of SynGAP1.

[1] Dementia Research Unit (DRU), School of Medical Sciences, The University of New South Wales, Sydney, NSW 2052, Australia. [2] Division of Cancer Genetics and Therapeutics, Laboratory of Chromatin, Epigenetics & Differentiation, Institute of Molecular and Cell Biology, A*STAR (Agency for Science, Technology and Research), Singapore 138673, Singapore. [3] The Jenner Institute, University of Oxford, Old Road Campus Research Building, Roosevelt Drive, Oxford OX3 7DQ, UK. [4] Translational Neuroscience Facility and Department of Physiology, School of Medical Sciences, The University of New South Wales, Sydney, NSW 2052, Australia. [5] Neuron Culture Core Facility (NCCF), The University of New South Wales, Sydney, NSW 2052, Australia. [6] Department of Biochemistry, Yong Loo Lin School of Medicine, National University of Singapore, Singapore 117549, Singapore. [7] Neurodegeneration and Repair Unit (NRU), School of Medical Sciences, The University of New South Wales, Sydney, NSW 2052, Australia. [8] Transgenic Animal Unit, Mark Wainwright Analytical Centre, The University of New South Wales, Sydney, NSW 2052, Australia. [9] Neuroscience Research Australia (NeuRA), Sydney, NSW 2031, Australia. Mian Bi, Amadeus Gladbach and Janet van Eersel contributed equally to this work. Yazi D. Ke and Lars M. Ittner jointly supervised this work. Correspondence and requests for materials should be addressed to L.M.I. (email: l.ittner@unsw.edu.au)

Stroke remains a major cause of disability and the second most common cause of death after cardiovascular conditions[1]. Ischemic strokes with acute focal brain infarction together with sudden and persisting neurological deficits are the most prevalent form. While neurons within the ischemic core region are likely to be irreversibly damaged, neurons in surrounding brain areas (referred to as the penumbra) are at risk of undergoing progressive necrotic/apoptotic death following the initial infarct[2]. There is only a short window for therapeutic intervention, aiming primarily at restoring blood flow to the ischemic brain areas either by pharmacological or mechanical

thrombolysis before neurons are irreversibly damaged[3–6]. However, the reperfusion itself may cause harm to neurons[2]. The mechanisms leading to neuronal damage following ischemia and reperfusion are multifaceted, including production of reactive oxygen species (ROS), mitochondrial failure and others[7]. A major contributor to neuronal damage in stroke is excitotoxicity[8], which results from over-excitation of glutaminergic synapses, particularly NMDA receptor (NMDAR) signaling[9]. However, many of its molecular pathways are yet to be identified.

The microtubule-associated protein tau is abundant in neurons, regulating stability and dynamics of microtubules[10]. It is

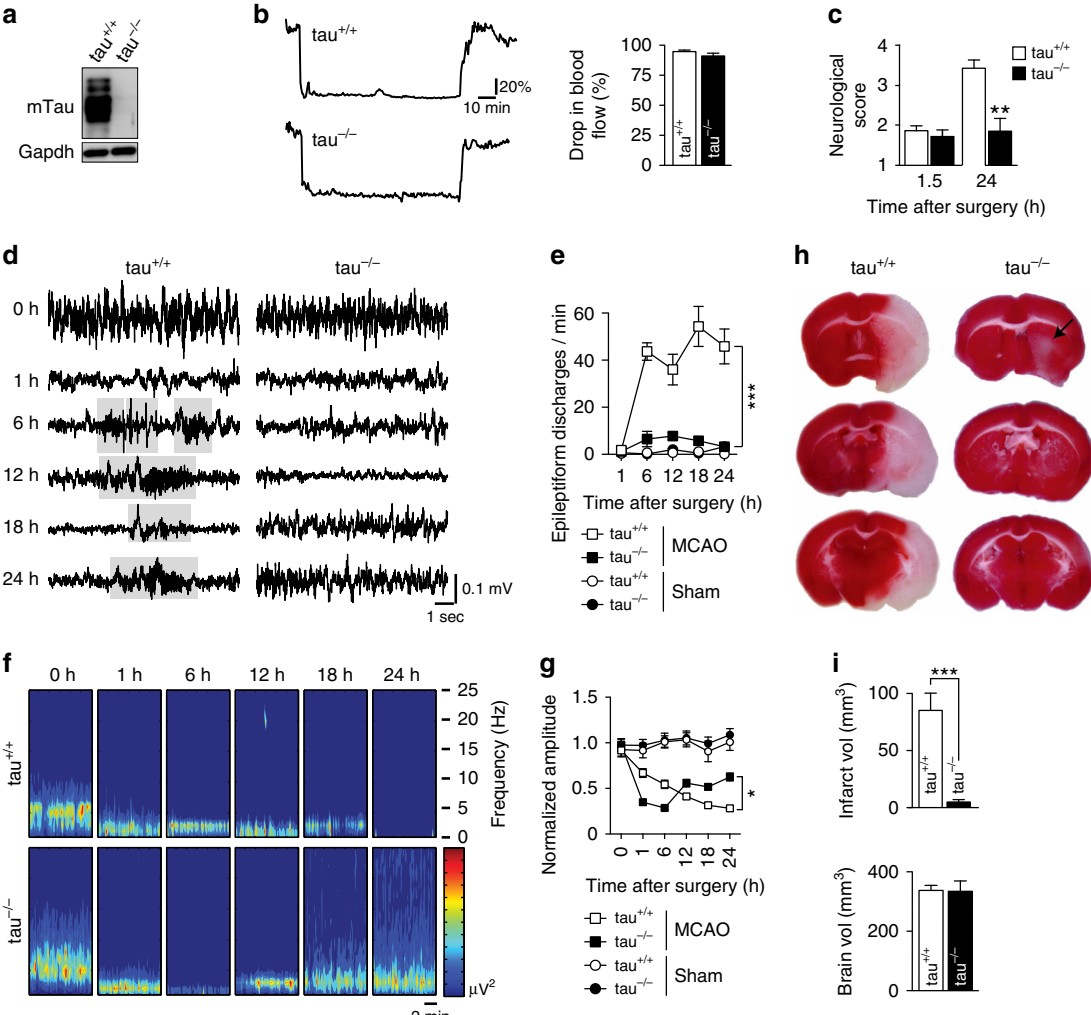

**Fig. 1** Tau$^{-/-}$ mice are protected from neurological deficits, aberrant hyperexcitation and extensive brain damage after transient MCAO. **a** Western blotting for murine tau (mTau) in brain extracts from tau$^{+/+}$ and tau$^{-/-}$ mice. GAPDH confirmed equal loading. **b** Ischemic stroke was induced by middle cerebral artery occlusion (MCAO) for 1.5 h with subsequent reperfusion. Drop in blood flow in the MCA was the same in tau$^{-/-}$ and tau$^{+/+}$ mice during MCAO, as determined by laser Doppler flowmetry (not significant; $N = 12$; Student's $t$-test). Unit, % of baseline flow. **c** Neurological scoring (with higher numbers indicating more severe impairments) revealed similar deficits directly after MCAO in tau$^{-/-}$ and tau$^{+/+}$ mice. Only tau$^{+/+}$ mice showed a worsening of deficits at 24 h (**$P < 0.01$; $N = 12$; 2-way ANOVA (Sidak post hoc)). **d** Representative electroencephalography (EEG) recordings in tau$^{-/-}$ and tau$^{+/+}$ mice at baseline (0 h) and indicated times after transient MCAO. Suppressed EEG signals recovered after 12 h following MCAO in tau$^{-/-}$ mice, while tau$^{+/+}$ mice presented with epileptiform discharges (*gray boxed*) after 6 h. **e** Quantification revealed persistently high numbers of epileptiform discharges 6 h after transient MCAO (***$p < 0.001$ vs. tau$^{-/-}$ MCAO; $N = 5$; two-way ANOVA (Bonferroni post hoc)). A small increase in epileptiform spike trains was transient in tau$^{-/-}$ mice, and reached levels of sham operated tau$^{-/-}$ and tau$^{+/+}$ mice 24 h after transient MCAO. **f** EEG frequency power spectrum (0–25 Hz) tau$^{+/+}$ and tau$^{-/-}$ mice at baseline (0 h) indicated times following transient MCAO, with recovery only in tau$^{-/-}$ animals. **g** Quantification of EEG recordings showed a progressive amplitude decline in tau$^{+/+}$ mice that partial recovered tau$^{-/-}$ animals (*$p < 0.05$; $N = 5$; two-way ANOVA (Bonferroni post hoc)). **h** TTC-stained serial brain sections of tau$^{+/+}$ and tau$^{-/-}$ mice 24 h after MCAO (viable tissue stains *red*). Note the large infarcted area (*white*) in tau$^{+/+}$, while only minimal brain damage was present in tau$^{-/-}$ mice (*arrow*). **i** Volumetric quantification of infarct and brain volumes in tau$^{+/+}$ and tau$^{-/-}$ mice 24 h after transient MCAO (***$p < 0.001$; $N = 12$; Student's $t$-test). All *error bars* are s.e.m

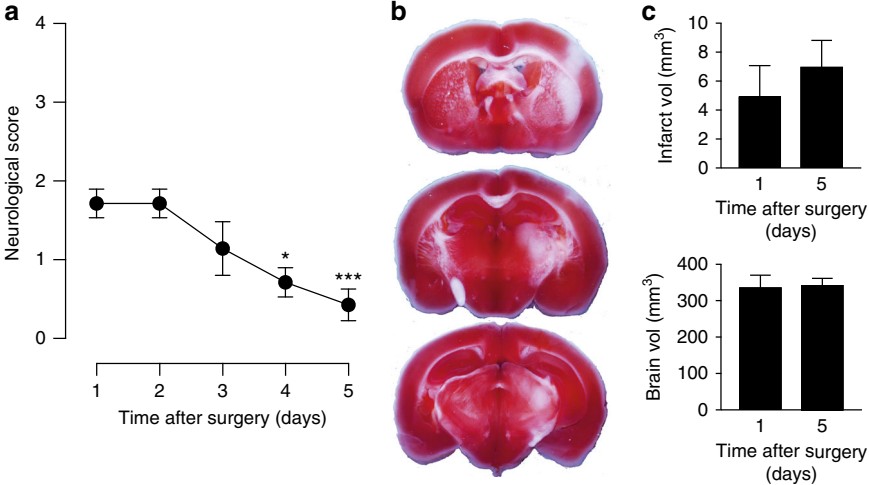

**Fig. 2** Improvement of functional deficits over 5 days after 90 min of transient MCAO in tau$^{-/-}$ mice. **a** Neurological impairments of tau$^{-/-}$ mice after 90 min of transient MCAO progressively improved over 5 days after the procedure (*$p < 0.05$; ***$p < 0.001$; $N = 8$; one-way ANOVA (Turkey post hoc)). **b** Example of a maximum infarct area 5 days after transient MCAO in tau$^{-/-}$ mice. TTC stains viable brain tissue in serial sections *red*. **c** Volumetric analysis showed similar infarct volumes 24 h and 5 days after MCAO (not significant; $N = 12$ (day 1), $N = 8$ (day 5); Student's *t*-test). All *error bars* are s.e.m

the major constituent of neurofibrillary tangles in Alzheimer's disease (AD) and frontotemporal dementia (FTD)[11]. Tau is progressively hyperphosphorylated in disease, which makes it prone to aggregation/deposition and interferes with its normal cellular functions[10, 12]. AD/FTD-like tau pathology has been reproduced in many mouse models by overexpressing tau, but interestingly tau$^{-/-}$ mice are phenotypically normal throughout development and adolescence, and may present deficits only at advanced ages[13]. Bearing similarities to early changes in AD, experimental animal models of stroke revealed changes in phosphorylation of tau, with reduction during early reperfusion after ischemia, followed by persisting hyperphosphorylation hours after the initial infarct[14–19]. Whether this reflects a general stress-response of neurons, or if tau plays a mechanistic role in stroke, however, remains unclear.

We and others have shown that memory deficits and early deaths in AD mice are tau-dependent[20–22]. Reducing tau in AD mouse models prevented excitotoxicity-mediated deficits, and tau-deficient mice showed protection from excitotoxic seizures[20, 21]. Given the role of excitotoxicity in stroke[23–25], we hypothesize that reduction of tau would reduce acute excitotoxic brain damage in stroke, which in turn would reveal a mechanistic role of tau in stroke. To test this hypothesis in vivo, we used tau-deficient mice together with models of experimental stroke and excitotoxicity. This approach revealed a profound protection from acute excitotoxic brain damage in the absence of tau, which is mediated, at least in parts, by site-specific inhibition of extracellular signal-regulated kinase (ERK) signaling.

## Results

**Tau$^{-/-}$ mice are protected from severe deficits after stroke.** To determine if tau contributes to brain damage following stroke, we subjected wild-type (tau$^{+/+}$) and tau$^{-/-}$ mice (Fig. 1a) to transient middle cerebral artery occlusion (MCAO) with reperfusion of ischemic brain areas, an experimental paradigm replicating clinical presentations of patients with successful recanalization or thrombolysis[26, 27]. We chose 90 min MCAO followed by reperfusion to produce infarcts[28] with profound and progressive expansion of brain damage over 24 h (h)[29].

Laser Doppler flowmetry confirmed MCAO and reperfusion (Fig. 1b). Neurological assessment after MCAO and recovery from anesthesia revealed comparable minor motor deficits in both tau$^{+/+}$ and tau$^{-/-}$ mice at reperfusion, indicating a similar degree of initial ischemic injury (Fig. 1c and Supplementary Fig. 1). Furthermore, blood parameters (pH, electrolytes, pCO$_2$, BE$_{ecf}$, HCO$_3$, total CO$_2$, Hct), body temperature, blood pressure, heart rate and O$_2$ saturation were similar in tau$^{+/+}$ and tau$^{-/-}$ mice before, during and 1 h after the procedure (Supplementary Table 1). There were also no overt differences in the vascular anatomy of the brain between tau$^{+/+}$ and tau$^{-/-}$ mice (Supplementary Fig. 2). Twenty four hours after MCAO, tau$^{+/+}$ mice developed profound neurological deficits (Fig. 1c), consistent with progressive brain damage. Strikingly, tau$^{-/-}$ mice did not develop further deficits within 24 h, suggesting protection from progression of transient MCAO-induced deficits.

Non-seizure epileptiform activity after focal cerebral ischemia has been reported in rodents[30, 31]. Therefore, we performed 24 h electroencephalography (EEG) recording using electrodes placed during the MCAO procedure to determine alterations in neuronal circuit response. In both tau$^{+/+}$ and tau$^{-/-}$ mice, baseline EEG recordings were comparable; as was the drop in EEG activity within 1 h after MCAO (Fig. 1d), further suggesting a similar degree of initial injury upon MCAO. Consistent with previous reports in rodents[30, 31], tau$^{+/+}$ mice experienced frequent epileptiform discharges from 6 h and throughout the 24 h recording period after MCAO (Fig. 1d, e). In contrast, tau$^{-/-}$ showed only very rare epileptiform discharges after MCAO, not significantly different from sham-operated mice. Furthermore, interictal total amplitude and EEG power across the frequency spectrum decreased progressively in tau$^{+/+}$ mice, while tau$^{-/-}$ mice showed gradual recovery after initial suppression of amplitude and spectral power (Fig. 1f, g), indicating functional protection of ipsilateral neuronal circuits. Taken together, EEG recording after transient MCAO revealed a similar level of initial depression of brain activity in tau$^{-/-}$ and tau$^{+/+}$ mice, but development of epileptiform activity consistent with hyperexcitation only occurred in tau$^{+/+}$ mice. Confirming that neuronal excitability was not compromised in tau$^{-/-}$ brains per se, we measured comparable Ca$^{2+}$ responses in brain slices of tau$^{+/+}$ and tau$^{-/-}$ mice expressing the GCaMP5G Ca$^{2+}$ reporter in neurons challenged with 1 mM glutamate (Supplementary Fig. 3).

Next, we determined if the protections in tau$^{-/-}$ mice was reflected by differences in infarct size compared to tau$^{+/+}$.

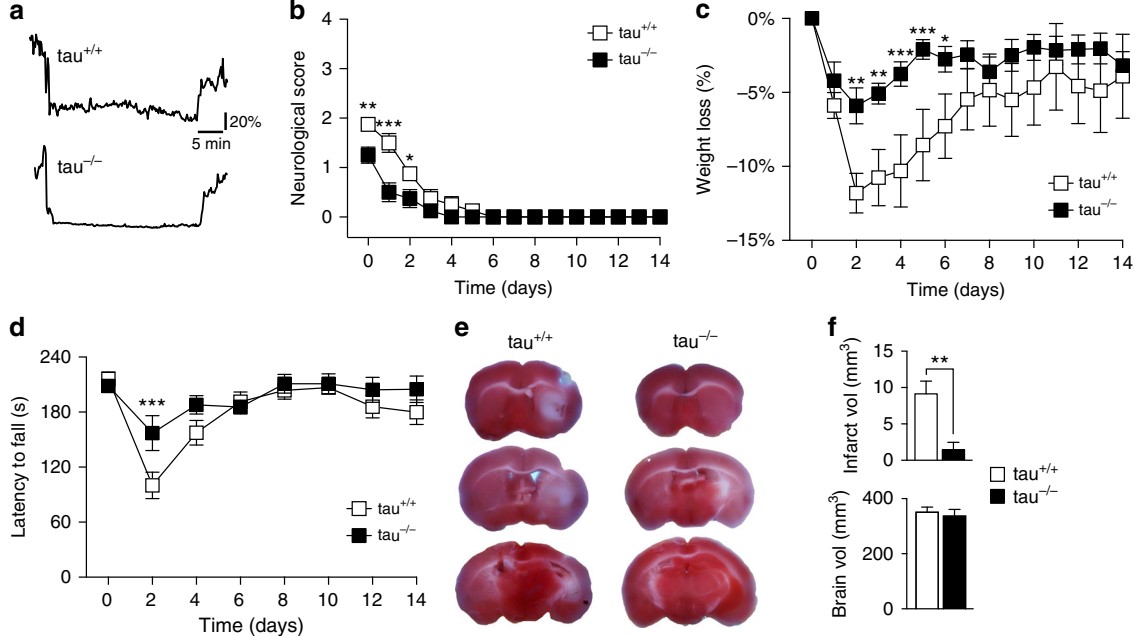

**Fig. 3** Accelerated recovery of tau$^{-/-}$ mice from transient MCAO. **a** For longer-term follow up of tau$^{-/-}$ and tau$^{+/+}$ mice after stroke, animals were exposed to 30 min of transient MCAO. Representative laser Doppler flowmetry of the MCAO procedure. Unit, % of baseline flow. **b** Significantly less neurological deficits with faster recovery in tau$^{-/-}$ compared to tau$^{+/+}$ mice following to 30 min of transient MCAO (*$p < 0.05$; **$p < 0.01$; ****$p < 0.0001$; $N = 8$; two-way ANOVA (Sidak post-hoc)). **c** Significantly less body weight loss with faster recovery in tau$^{-/-}$ compared to tau$^{+/+}$ mice following to 30 min of transient MCAO (*$p < 0.05$; **$p < 0.01$; ***$p < 0.001$; $N = 8$; two-way ANOVA (Sidak post hoc)). **d** Tau$^{-/-}$ mice performed significantly better on the accelerative mode Rota-Rod than tau$^{+/+}$ animals (***$p < 0.001$; $N = 8$; two-way ANOVA (Sidak post hoc)). **e** TTC-stained serial brain sections of tau$^{+/+}$ and tau$^{-/-}$ mice 14 days after 30 min of transient MCAO (viable tissue stains *red*). Note the larger infarcted brain area (*white*) in tau$^{+/+}$, compared to minimal brain damage in tau$^{-/-}$ mice. **f** Quantification of infarct and brains volumes in tau$^{-/-}$ and tau$^{+/+}$ mice (**$P < 0.001$; $N = 8$; Student's *t*-test). All *error bars* are s.e.m

As expected, 24 h after transient MCAO, tau$^{+/+}$ brains showed pronounced infarcts (Fig. 1h). In contrast, tau$^{-/-}$ brains had drastically smaller lesions compared to tau$^{+/+}$ mice (Fig. 1i). Early loss of neuronal microtubule-associated protein 2 (MAP2) staining has been reported after stroke[32]. Accordingly, MAP2 staining was profoundly and broadly reduced in the ischemic cortex of tau$^{+/+}$ mice 3 h after transient MCAO, while a moderate reduction of MAP2 staining in tau$^{-/-}$ brains was confined to a small core area without changes in areas of the cortex that corresponded to those affected in tau$^{+/+}$ mice (Supplementary Fig. 4). Tau$^{-/-}$ mice observed for up to 5 days after transient MCAO did show progressive improvement of the mild neurological deficits with no increase in infarct volumes (Fig. 2a–c), suggesting persisting protection. Taken together, reduced neurological deficits after 90 min of transient MCAO in tau$^{-/-}$ mice were associated with markedly smaller infarcts.

The substantial brain damage and profound functional deficits in tau$^{+/+}$ mice after 90 min of MCAO did not allow following up tau$^{+/+}$ mice for longer than 24 h. Therefore, we subjected an additional cohort of tau$^{+/+}$ and tau$^{-/-}$ mice to 30 min of MCAO allowing longer-term follow up experiments (Fig. 3a). Following this milder transient MCAO, tau$^{-/-}$ mice displayed significantly less severe neurological deficits compared to tau$^{+/+}$ mice receiving this treatment (Fig. 3b). Both improved their functional deficits over the following days, with tau$^{-/-}$ displaying no neurological deficits on day 4 after MCAO, while tau$^{+/+}$ mice required 6 days to fully recover. Similarly, tau$^{-/-}$ mice lost significantly less body weight within 2 days after MCAO, and recovered the lost weight faster than tau$^{+/+}$ animals (Fig. 3c). Notably, tau$^{+/+}$ did not fully recover the lost body weight within 2 weeks after 30 min of transient MCAO. Both, tau$^{+/+}$ and tau$^{-/-}$ mice showed significantly decreased performance on the accelerating Rota-

Rod 2 days after the MCAO procedure, although tau$^{-/-}$ mice performed significantly better than tau$^{+/+}$ mice (Fig. 3d). These deficits recovered fully 4 and 6 days after MCAO in tau$^{-/-}$ and tau$^{+/+}$ mice, respectively. As expected, the brain damage was substantially less in tau$^{-/-}$ than tau$^{+/+}$ mice 14 days after 30 min of transient MCAO (Fig. 3e, f). Hence, reduced neurological and functional deficits were less profound and recovery was faster in tau$^{-/-}$ than tau$^{+/+}$ mice after 30 min of transient MCAO, and was associated with markedly smaller infarcts.

**Mitigation of excitotoxic gene response in tau$^{-/-}$ mice.** Neuronal hyperexcitation results in expression of immediate-early genes (IEGs), including Arc[33], cFos and Junb[34]. Accordingly, Arc, cFos and Junb mRNA levels were $5.3 \pm 1.4$-fold, $5.4 \pm 1.9$-fold, and $4.0 \pm 0.7$-fold ($p < 0.001$; $N = 6$) higher in the ipsilateral hemisphere of tau$^{+/+}$ brains 1 h after transient MCAO (Fig. 4a). In contrast, induction of IEG transcription was blunted in the corresponding brain region in tau$^{-/-}$ mice (Fig. 4a). Consistent with differential Arc mRNA regulation, increased staining for neuronal Arc protein in tau$^{+/+}$, but not tau$^{-/-}$ brains was observed 3 h after transient MCAO (Supplementary Fig. 5a). Furthermore, phosphorylated (p) H2AX staining, indicating cell damage with DNA breaks[35], was restricted to the ischemic core in tau$^{-/-}$ brains, while being widely detectable in tau$^{+/+}$ brains 3 h after MCAO (Supplementary Fig. 5b). Hence, absence of a pronounced IEG expression and cell damage during the initial phase following MCAO, suggest protection of tau$^{-/-}$ neurons from early toxic signaling events after re-perfusion.

Using the PTZ model allowed us to isolate excitotoxic processes under tightly controlled experimental conditions in tau$^{+/+}$ and tau$^{-/-}$ mice, utilizing it as a screening model before

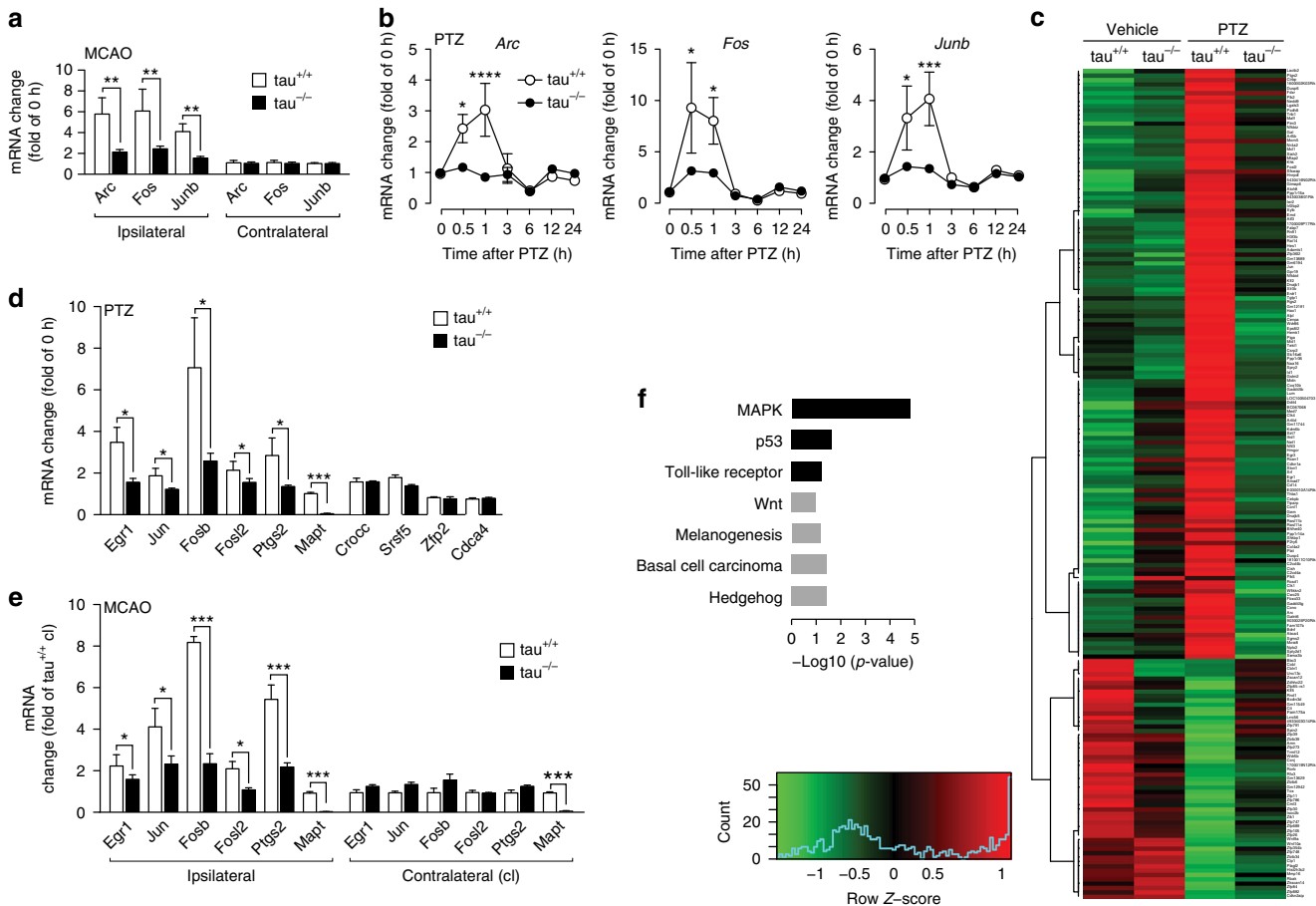

**Fig. 4** Excitotoxic seizures and transient MCAO cause similar differential gene regulation in tau$^{-/-}$ mice. **a** Immediate early gene (IEG) activation for Arc, Fos and Junb in the affected hemisphere (ipsilateral) 1 h after transient MCAO in tau$^{+/+}$ and tau$^{-/-}$ mice (**$p < 0.01$; $N = 6$; Student's $t$-test). No gene induction occurred on the contralateral side. **b** Time course of IEG mRNA induction of Arc, Fos and Junb after PTZ administration in tau$^{+/+}$ but little to none in tau$^{-/-}$ mice (*$p < 0.05$; ***$p < 0.001$; ****$p < 0.0001$; $N = 6$ per time point; two-way ANOVA (Holm-Sidak post hoc)). mRNA levels were determined by real-time PCR (rtPCR) using gene-specific primers listed in Supplementary Table 3. **c** Whole transcriptome sequencing of tau$^{-/-}$ and tau$^{+/+}$ mice treated with vehicle or PTZ revealed pronounced lack of gene induction/suppression 1 h after treatment in tau$^{-/-}$ mice. *Red* indicates up- and *green* down-regulation. Only genes with significant differential regulation are displayed. The displayed gene names are listed in order in Supplementary Data 1. **d** Quantitative rtPCR confirms differential regulation of selected transcripts from (see **c**) tau$^{-/-}$ mice (*$p < 0.05$; ***$p < 0.0001$; $N = 6$; Student's $t$-test), as well as similar regulation of others (selected from Supplementary Fig. 6 and Supplementary Data 2) in tau$^{-/-}$ and tau$^{+/+}$ mice. **e** Quantitative rtPCR showed a similar pattern of differentially regulated genes in the ipsilateral hemisphere after transient MCAO compared to after PTZ (see **d**) (*$p < 0.05$; ***$p < 0.0001$; $N = 6$; Student's $t$-test), while there was no deregulation contralaterally in tau$^{-/-}$ and tau$^{+/+}$ mice. **f** DAVID pathway analysis of genes that lacked response to PTZ administration in tau$^{-/-}$ mice (see **c**) identified pathways that were up- (*black*) and down-regulated (*gray*) in tau$^{+/+}$ mice. All *error bars* are s.e.m

translating back into the more complex MCAO model. Similar to our findings following transient MCAO, excitotoxicity-associated IEGs mRNA levels of Arc, cFos and Junb were markedly increased in tau$^{+/+}$, but not tau$^{-/-}$ mice in the PTZ-induced excitotoxic seizure model 0.5 and 1 h after administration (Fig. 4b). Using this experimental paradigm, we employed whole transcriptome sequencing to determine if absence of excitotoxicity-associated IEGs in tau$^{-/-}$ mice reflects a general lack of response, or if only distinct genes are differentially regulated. Global comparison of mRNA levels in brains 1 h after administration of PTZ showed clusters of both differentially and similarly regulated genes in tau$^{+/+}$ and tau$^{-/-}$ mice (Fig. 4c, Supplementary Fig. 6 and Supplementary Data 1 and 2). Quantitative PCR of selected genes independently confirmed this differential regulation upon PTZ administration (Fig. 4d), and shows a similar profile of differentially regulated genes following MCAO of tau$^{+/+}$ and tau$^{-/-}$ mice (Fig. 4e). Taken together, differential regulation of gene clusters suggests differential

activation of distinct signaling pathways in tau$^{+/+}$ and tau$^{-/-}$ during excitotoxicity upon PTZ administration and MCAO.

Given the prominent role of excitotoxicity in stroke[8] and that tau$^{-/-}$ mice show an increased latency to developing severe excitotoxic seizures induced by 50 mg/kg pentylenetetrazole (PTZ)[20, 21], we focused our mechanistic investigation on the contribution of excitotoxic signaling to brain damage. Focusing on excitotoxicity was further supported by larger brain damage after intra-cortical infusion of NMDA in tau$^{+/+}$ compared to tau$^{-/-}$ mice (Fig. 5a, b), and reduced NMDA-mediated neuronal death in tau$^{-/-}$ compared with tau$^{+/+}$ primary cultured neurons (Fig. 5c, d).

**Tau$^{-/-}$ mice lack excitotoxic Ras/ERK activation.** DAVID pathway analysis indicated MAPK, p53, toll-like receptor, Wnt, melanogenesis, basal cell carcinoma and hedgehog signaling being differentially engaged upon PTZ-administration in tau$^{+/+}$

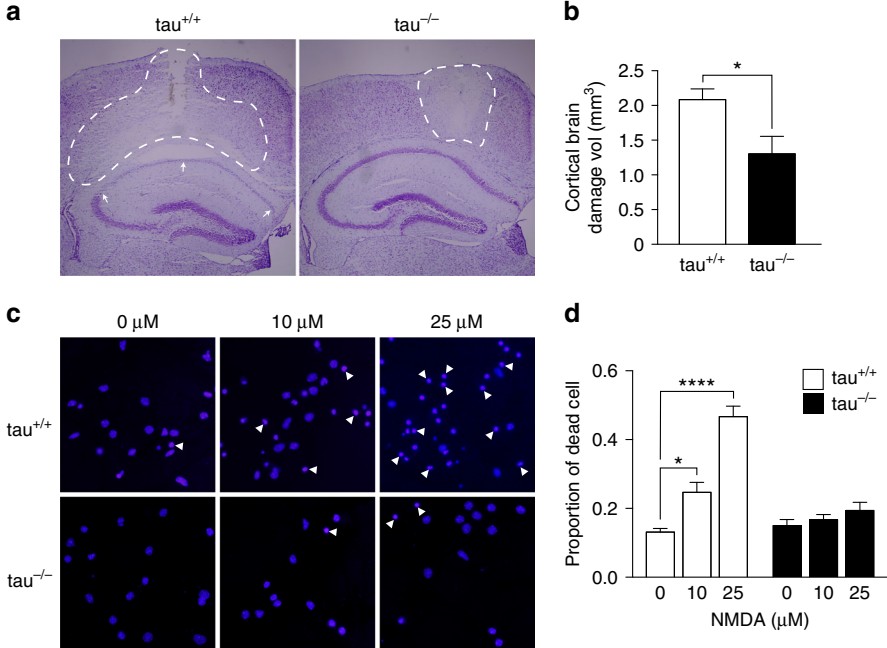

**Fig. 5** Tau-depletion reduced neurotoxicity of NMDA. **a** Representative Nissl staining of tau[+/+] and tau[−/−] brains 24 h after infusion of 0.2 μl of 50 mM NMDA into the cortex with pale appearance of damaged brain area (*broken lines*). Note that the damaged area extends profoundly into the CA1 region of the hippocampus (*arrows*) in tau[+/+] but not tau[−/−] brains, suggesting progressive expansion of excitotoxicity only in tau[+/+] brains. **b** Quantification confirmed less total brain damage in tau[−/−] compared to tau[+/+] mice (*$p < 0.05$, N = 4; Student's t-test). **c** Primary cortical neurons treated with 0, 10 and 25 μM NMDA showed less cell death (determined by nuclear condensation (some indicated by *arrowheads*), DAPI) in tau[−/−] compared with tau[+/+] cells. **d** Quantification of neuron death induced by NMDA in tau[−/−] and tau[+/+] cells (*$p < 0.05$; ****$p < 0.0001$; N = 12; two-way ANOVA (Sidak post hoc)). All *error bars* are s.e.m

and tau[−/−] mice (Fig. 4f and Supplementary Table 2). Amongst these, MAPK signaling appeared to be most significantly affected in tau[−/−] mice (Fig. 4f), with the majority of differentially regulated target genes being down-stream of ERK1/2 (Supplementary Table 2). Therefore, we next investigated ERK activation in tau[+/+] and tau[−/−] brains after PTZ administration. In tau[+/+] mice, PTZ induced transient but pronounced ERK phosphorylation 10 min after administration that returned to baseline within 30 min (Fig. 6a, b). In contrast, virtually no increase in ERK phosphorylation occurred in brains of PTZ-treated tau[−/−] mice. To directly assess NMDAR-mediated activation of ERK signaling, we treated primary neurons with NMDA (Fig. 6c, d). This resulted in increased levels of ERK phosphorylation in tau[+/+] neurons, but no activation and rather decreased phosphorylation of ERK in tau[−/−] cells. This is in line with a failure to activate ERK downstream of synaptic NMDARs, while ERK inhibition mediated by extra-synaptic NMDARs[36] remained intact in tau[−/−] neurons. Accordingly, treatment of primary neurons with bicuculline or KCl to increase synaptic glutamate levels and induce excitotoxic signaling[37, 38] resulted in ERK phosphorylation and IEGs induction in tau[+/+] but not tau[−/−] cells (Fig. 6e–g). Significant ERK phosphorylation upon receptor-independent activation with forskolin in tau[+/+] and tau[−/−] neurons suggests that the ERK signaling cascade per se is functional, but at a reduced level in tau[−/−] mice (Fig. 6e, f), indicating some form of inhibition. Taken together, tau[−/−] mice lack ERK activation upon excitotoxic stimulation.

NMDARs activate ERK via the Ras/Raf/MEK signaling pathway[39]. We have previously shown that tau-depletion compromises NMDAR downstream signaling[21]. Despite tau interacting with PSD-95[21, 40], its role in down-stream excitotoxic signaling remained to be shown. To identify the step of the Ras/Raf/MEK cascade in which tau regulates NMDAR-induced

ERK activation, we first quantified excitotoxic Ras activation by Raf-mediated pull-down of active, GTP-bound Ras. Surprisingly, no Ras activation occurred in tau[−/−] neurons after hyperexcitation compared to tau[+/+] (Fig. 6h, i), despite indistinguishable total and synaptic Ras levels (Fig. 6h and Supplementary Fig. 7), suggesting tau regulation of ERK activation is upstream of Ras.

RasGRF, a guanine-nucleotide exchange factor that converts GDP-bound Ras into its active GTP-bound state, has previously been shown to mediate Ras activation down-stream of NMDAR[39, 41]. In contrast, SynGAP1 is an endogenous inhibitor of Ras at the post-synapse[42, 43] that catalyzes GTP hydrolysis thereby inactivating Ras[44]. We first tested for changes in total and synaptic levels of RasGRF in tau[−/−], but did not find any alterations (Supplementary Fig. 7). Similarly, the interaction between RasGRF and NMDARs was unchanged in tau[−/−] compared to tau[+/+] mice, and we found no evidence of an interaction between RasGRF and tau that may have indicated a functional relevance possibly lost in tau[−/−] mice (Supplementary Fig. 7). In contrast, immunofluorescence staining revealed significantly stronger labeling of SynGAP1 that co-localizes with post-synaptic PSD-95 in tau[−/−] compared to tau[+/+] neurons, indicative of increased synaptic levels of SynGAP1 in tau[−/−] neurons (Fig. 6j, k). Since SynGAP1 resides at the post-synaptic density (PSD)[42, 43], we next determined if the functional interaction between SynGAP1 and PSD-95 is altered in the absence of tau. To our surprise, immunoprecipitation of SynGAP1 co-purified markedly more PSD-95 from tau[−/−] than tau[+/+] brains (Fig. 6l, m), indicating that more SynGAP1 is bound to PSD-95 in the absence of tau. This raised the question if tau has a direct regulating effect on post-synaptic SynGAP1. Therefore, we first assessed if tau and SynGAP1 reside together in a complex by co-immunoprecipitation. Immunoprecipitation with 3 different tau antibodies co-precipitated SynGAP1 from tau[+/+]

brain extracts (Fig. 6n and Supplementary Fig. 7). Since co-immunoprecipitation from tissue lysates does not provide spatial information, we next employed antibody-based fluorescence in situ proximity determination in primary $tau^{+/+}$ neurons. Using tau- and SynGAP1-specific antibodies, we found that tau and SynGAP1 reside in close proximity exclusively within dendritic spines of $tau^{+/+}$ neurons (Fig. 6o). Finally,

SynGAP1 co-immunoprecipitate with PSD-95 in the absence, but not in the presence of co-expressed tau from transiently transfected cells (Fig. 6p), suggesting negative regulation of PSD-95/SynGAP1 complexes by tau. In summary, our data suggests a post-synaptic accumulation of SynGAP1 in $tau^{-/-}$ mice, where it blocks Ras activation and therefore NMDAR-mediated ERK signaling.

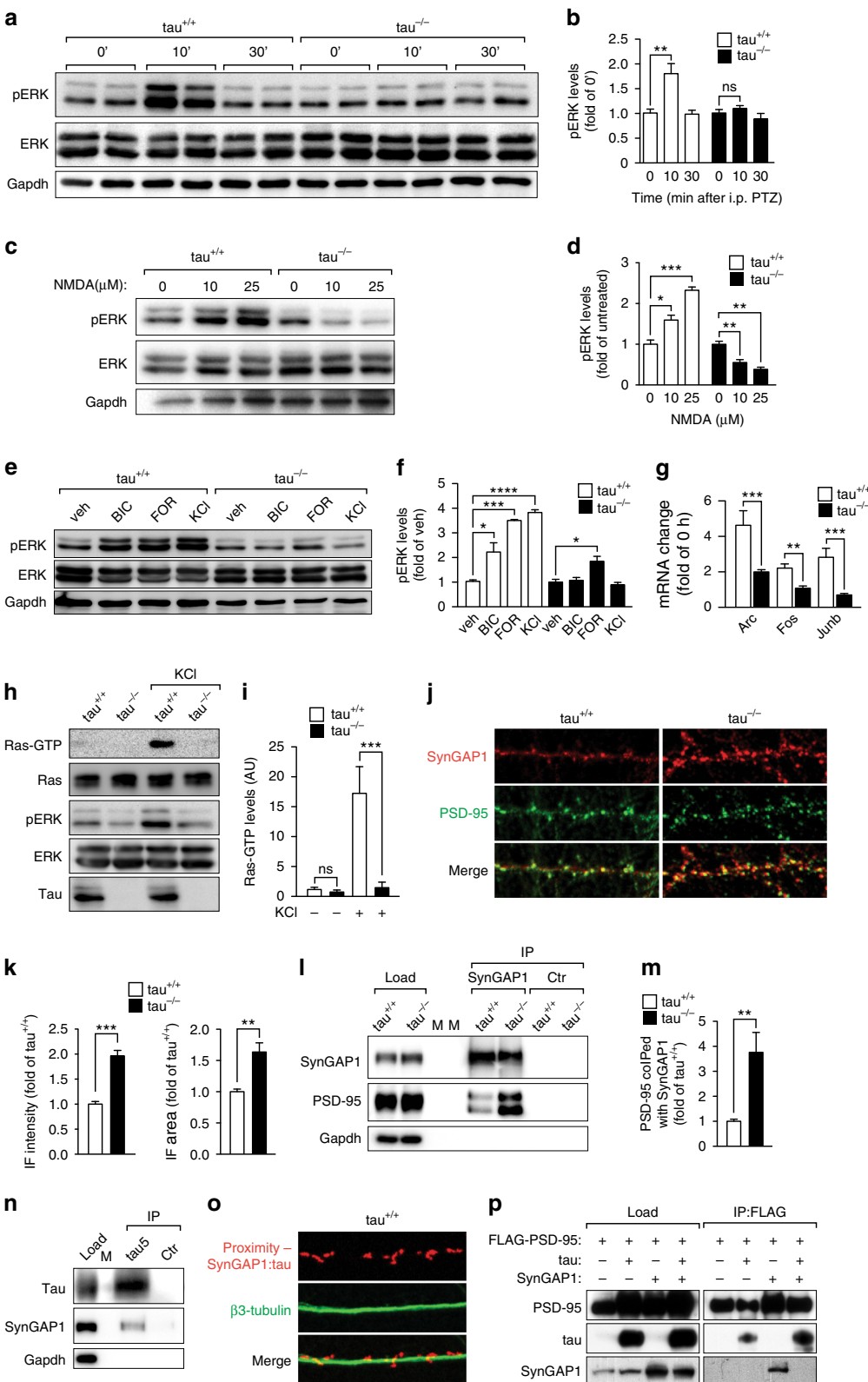

**Reducing SynGAP1 restores susceptibility in tau$^{-/-}$ mice.** If increased post-synaptic accumulation of SynGAP1 in tau$^{-/-}$ neurons contributes to the protection from excitotoxicity, reducing SynGAP1 levels should render tau$^{-/-}$ neurons again susceptible to excitotoxic damage and possibly brain damage after MCAO. Therefore, we designed short hairpin RNA (shRNA) for AAV-mediated targeted knockdown of SynGAP1 (AAV-SG1-shR) in neurons of tau$^{-/-}$ mice. We administered AAV-SG1-shR or a control shRNA-expressing AAV (AAV-ctr-shR) by intra-cranial injections at postnatal day P1-3 in mice and showed significantly reduced levels of SynGAP1 in AAV-SG1-shR-injected, compared to AAV-ctr-shR-injected and naive mice at 2 months of age (Fig. 7a). Identification of transduced cells was determined by bicistronic expression of a GFP reporter cassette downstream of the shRNA. Imaging showed the widespread distribution of GFP expression throughout the brain and reduction of SynGAP1 staining in transduced areas (Fig. 7b). Hence, intracranial AAV-SG1-shR delivery efficiently reduced SynGAP1 levels in vivo.

We then administered AAV-SG1-shR or AAV-ctr-shR in newborn tau$^{-/-}$ mice to determine the effects of SynGAP1 reduction in vivo. First, we induced excitotoxic seizures by administering 50 mg/kg PTZ to 2 month-old tau$^{-/-}$ mice, which had received intracranial injections of either AAV-SG1-shR or AAV-ctr-shR at P1-3. Both mean seizure severity and latency to developing seizures were comparable in AAV-ctr-shR-injected and naive tau$^{-/-}$ mice (Fig. 7c, d), suggesting no effects from AAV-injection and shRNA expression per se. In contrast, AAV-SG1-shR injections increased the mean seizure severity of tau$^{-/-}$ mice significantly. While the latency was not significantly reduced in AAV-SG1-shR-injected tau$^{-/-}$ mice, more mice developed severe bouncing seizures that were rarely seen in AAV-ctr-shR-injected or naive tau$^{-/-}$ mice within 10 min after PTZ administration (Fig. 7d). In parallel to restoring sensitivity of tau$^{-/-}$ mice to PTZ-induced seizures by knocking down SynGAP1, Western blotting revealed that activation of ERK signaling in response to PTZ administration was similar in tau$^{-/-}$ mice injected with AAV-SG1-shR and tau$^{+/+}$ mice (Figs. 7e and 6a). In tau$^{+/+}$ mice, there was a moderate trend to a reduced latency to develop more severe seizures with more animals progressing to convulsive seizures when injected with AAV-SG1-shR compared with AAV-ctr-shR-injected controls (Supplementary Fig. 8). Correspondingly, AAV-SG1-shR-injected

tau$^{+/+}$ mice showed a trend towards higher levels of ERK phosphorylation 10 min after PTZ compared to AAV-ctr-shR-injected controls (Supplementary Fig. 8). Conversely, SynGAP1 over-expression in primary neurons mitigated NMDA-induced ERK phosphorylation (Fig. 8a, b).

Finally, we performed transient 1.5 h MCAO with reperfusion in tau$^{-/-}$ mice that were previously injected with AAV-SG1-shR or AAV-ctr-shR at P1-3 (Fig. 9a). AAV-ctr-shR-injected tau$^{-/-}$ mice showed the same minor neurological deficits (Fig. 9b) as naive tau$^{-/-}$ mice after transient MCAO (Fig. 1c). In contrast, knockdown of SynGAP1 in tau$^{-/-}$ mice was associated with severe neurological deficits 24 h after MCAO (Fig. 9b), similar to tau$^{+/+}$ mice (Fig. 1c). The neurological deficits were paralleled by large MCAO-induced brain infarcts in AAV-SG1-shR-injected tau$^{-/-}$ mice after 24 h, while the infarct sizes in AAV-ctr-shR-injected tau$^{-/-}$ mice remained small (Fig. 9c, d). Infarct sizes were comparable 24 h after transient MCAO in in AAV-SG1-shR-injected tau$^{+/+}$ mice compared to AAV-ctr-shR-injected controls, and their neurological deficits did not worsen any further, likely due to the already profound deficits of controls (Supplementary Fig. 8). Taken together, reducing neuronal SynGAP1 levels in tau$^{-/-}$ mice re-established their sensitivity to excitotoxic injury and transient MCAO-induced deficits and brain damage, suggesting tau-mediated excitotoxicity involves control of post-synaptic SynGAP1.

## Discussion

In the present study, we show that genetic depletion of tau prevents brain damage and neurological deficits after MCAO-induced stroke in mice. In parallel, tau$^{-/-}$ mice lack pronounced excitotoxic IEG response and ERK activation. Mechanistically, we have shown that tau limits the binding of SynGAP1, a site-specific inhibitor of Ras, to the post-synaptic PSD-95 protein complex, enabling Ras-mediated ERK activation downstream of NMDARs (Fig. 10).

Although the level of protection from brain damage after stroke in tau$^{-/-}$ mice is substantial, it is not unprecedented; a similar ~90% reduction of brain damage after 90 min MCAO has for example been achieved when targeting the NMDAR/PSD-95 complex with interfering peptides[24, 45] or blocking ERK signaling[46]. Interestingly, the molecular target that conferred this substantial protection[24] is within the same post-synaptic pathway

**Fig. 6** SynGAP1 accumulation blocks Ras/ERK signaling at the post-synapse in tau$^{-/-}$ mice. **a** Western blotting of brain extracts; PTZ induced transient ERK phosphorylation 10 min after administration in tau$^{+/+}$ but not tau$^{-/-}$ mice, **b** as confirmed by quantification of independent blots (\*\*$p < 0.01$; $N = 6$; one-way ANOVA (Tukey post hoc)). **c** Primary tau$^{+/+}$ but not tau$^{-/-}$ neurons showed ERK phosphorylation when challenged with 10 and 25 μM NMDA, **d** as confirmed by quantification of independent blots (\*$p < 0.05$; \*\*$p < 0.01$; \*\*\*$p < 0.001$; $N = 3$; one-way ANOVA (Tukey post hoc)). **e** Primary tau$^{+/+}$ neurons showed ERK phosphorylation when treated with bicuculline (BIC), forskolin (FOR) and KCl. In contrast, primary tau$^{-/-}$ neurons showed only increased ERK phosphorylation upon treatment with FOR, though to a lower degree. **f** Quantification of independent blots confirmed BIC-, FOR- and KCl-induced ERK phosphorylation in tau$^{+/+}$ neurons, while only FOR significantly activated ERK in tau$^{-/-}$ cells (\*$p < 0.05$; \*\*\*$p < 0.001$; \*\*\*\*$p < 0.0001$; $N = 6$; one-way ANOVA (Tukey post hoc)). **g** BIC treatment of primary neurons resulted in upregulation of Arc, Fos and Junb mRNA levels in tau$^{+/+}$, but little to none in tau$^{-/-}$ cells (\*\*$p < 0.01$; \*\*\*$p < 0.001$; $N = 6$; Student's $t$-test). **h** Activated Ras (Ras-GTP) was pulled down from KCl treated tau$^{+/+}$, but not tau$^{-/-}$ brain slices. Total Ras levels, however, were comparable tau$^{-/-}$ and tau$^{+/+}$ mice. Concomitant ERK activation was only seen in KCl treated tau$^{+/+}$ slices. **i** Quantification of independent blots confirmed that Ras activation occurred only in tau$^{+/+}$ mice (\*\*\*$p < 0.001$; $N = 6$; one-way ANOVA (Tukey post hoc)). **j** Confocal imaging of primary neurons co-stained with SynGAP1 (*green*) and PSD-95 (*red*) showed more intensive labeling of spines for SynGAP1 that co-localized with PSD-95 in tau$^{-/-}$ than in tau$^{+/+}$ neurons. **k** Both, immunofluorescence intensity and cluster size of SynGAP1 in spines were significantly increased in tau$^{-/-}$ compared to tau$^{+/+}$ neurons (\*\*$p < 0.01$; \*\*\*$p < 0.001$; $N = 5$; Student's $t$-test). **l** Immunoprecipitation (IP) with a SynGAP1-specific antibody co-purified 3.8-fold more PSD-95 from tau$^{-/-}$ brain extracts than from tau$^{+/+}$ lysates. Control (ctr) precipitations were done without primary antibodies. Load represents brain extracts used for immunoprecipitation. M, marker. **m** Quantification of 6 independent experiments showed a significant increase in SynGAP1/PSD-95 interaction in tau$^{-/-}$ compared to tau$^{+/+}$ mice (\*\*$p < 0.01$; $N = 6$; Student's $t$-test). **n** SynGAP1 interacts with tau in brain extracts from wild-type mice, as revealed by co-IP using a tau-specific antibodies (tau5). Control (ctr) precipitations were done without primary antibodies. M, marker lanes. **o** SynGAP1 and tau are in a complex within dendritic spines, as determined by a < 40 nm proximity ligation assay (SynGAP1:tau; *red*) using SynGAP1- and tau-specific antibodies. Dendritic shafts were counter-stained with β3-tubulin (*green*). **p** IP of FLAG-PSD95 co-precipitated SynGAP1 in the absence but not in the presence of tau from transiently transfected cells (load) ($N = 3$). All *error bars* are s.e.m

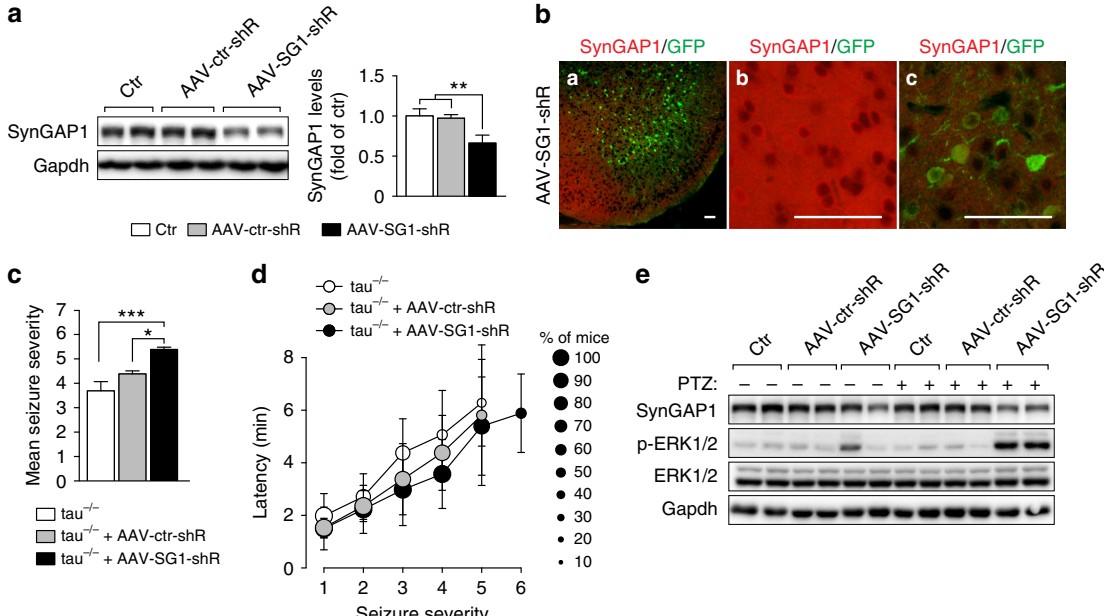

**Fig. 7** Reducing SynGAP1 levels in tau$^{-/-}$ mice reinstates susceptibility to seizures and ERK activation. **a** Neonatal (P1-3) wild-type C57Bl/6 mice were injected intracranially with AAVs expressing either shRNA to knockdown SynGAP1 (AAV-SG1-shR) or control shRNA (AAV-ctr-shR). Western blotting of cortical extracts from 2-month-old naive (ctr) and AAV-injected mice showed marked reduction of SynGAP1 levels in AAV-SG1-shR-injected mice. Quantification of band intensities confirmed significant SynGAP1 reduction by AAV-SG1-shR (**$p < 0.01$; $N = 9$; one-way ANOVA (Tukey post hoc)). **b** Example of GFP reporter expression in neurons throughout the cortex of AAV-SG1-shR-injected mice. Double labeling with a SynGAP1 antibody (*red*) revealed SynGAP1 expression in areas with no GFP expression and marked reduction of SynGAP1 in GFP-positive areas, suggesting efficient knockdown. Higher magnification of areas with neuronal and synaptic SynGAP1, but no GFP expression (*middle*), or neuronal GFP reporter expression with reduced SynGAP1 (*right*). *Scale bars*, 100 μm. **c** AAV-SG1-shR, but not AAV-ctr-shR injection increased the mean seizure severity in tau$^{-/-}$ mice after administration of 50 mg/kg PTZ (*$p < 0.05$; ***$p < 0.001$; $N = 11$ (tau$^{-/-}$), $N = 13$ (AAV injected tau$^{-/-}$); one-way ANOVA (Tukey post hoc)). **d** More tau$^{-/-}$ mice injected with AAV-SG1-shR developed higher degree seizures after administration of 50 mg/kg PTZ, compared with naive (ctr) or AAV-ctr-shR-injected tau$^{-/-}$ mice. The latency of seizure progression was comparable. **e** Strong ERK1/2 phosphorylation 10 min after 50 mg/kg PTZ administration in AAV-SG1-shR-injected, but not AAV-ctr-shR-injected or naive (ctr) tau$^{-/-}$ mice, together with reduction of SynGAP1 levels. Detection of ERK1/2 and Gapdh confirmed equal loading. All *error bars* are s.e.m

complex we show in the present study in tau$^{-/-}$ mice. Liu et al.[29] showed that 90 min MCAO caused a small area of brain damage within 1 h after the procedure, that gradually increased in size involving a majority of the hemisphere 24 h later, far beyond the terminal supply area of the MCA. For comparison, the brain damage in tau$^{-/-}$ mice 24 h after MCAO was of a size similar to that reported for wild-type mice directly after MCAO[26, 29]. Importantly, the reduction in infarct size in tau$^{-/-}$ mice was despite similar initial functional deficits, reduction in EEG activity and physiological responses directly following the procedure compared to tau$^{+/+}$ littermates. Furthermore, neuronal excitability in tau$^{-/-}$ and tau$^{+/+}$ mice indistinguishable. Our data suggests that tau significantly contributes to progressive damage of at-risk brain areas, likely by regulating excitotoxic signaling. Differences in brain vasculature, however, are unlikely to underlie the protection from stroke in tau$^{-/-}$ mice, since laser Doppler flowmetry showed the same reduction in cerebral blood flow during the MCAO procedure in tau$^{+/+}$ and tau$^{-/-}$ mice, and reduction of SynGAP1 levels restored susceptibility to brain damage after MCAO in tau$^{-/-}$ mice. The latter molecular intervention would not affect cerebral blood supply and would not have abolished the protection in tau$^{-/-}$ mice if vascular differences conferred this protection.

Others and we have previously reported that tau-deficient mice are protected from Aβ-induced deficits in AD mouse models, due to reduced susceptibility to excitotoxic neuronal damage[20, 21, 47]. The protection of tau$^{-/-}$ (and to a lesser degree tau$^{+/-}$) mice from excitotoxic brain damage received further support using either

pharmacological[20, 21] or genetic epilepsy models[48], as well as by direct intra-cortical NMDA infusion used in our study. Brain damage due to ischemia is orchestrated by a range of molecular mechanisms, including excitotoxicity[8]. Using the PTZ model of a temporally controlled and synchronized excitotoxic response, allowed us to explore the distinct tau-dependent molecular mechanisms. Interestingly, the cellular response to both the MCAO and PTZ paradigms were remarkably similar at the gene regulation level (Fig. 4). Furthermore, translating the molecular mechanisms of tau-dependent SynGAP1 regulation at the post-synapse back into the MCAO model supported our approach.

Differential mRNA expression in PTZ-treated tau$^{-/-}$ mice together with pathway prediction suggested compromised MAPK signaling. Combining experiments in primary neurons and in vivo, we were able to show that tau is required to induce IEG response and mediate cell damage via Ras/ERK signaling down-stream of NMDARs hyperexcitation. Supporting that a lack of Ras/ERK signaling in tau$^{-/-}$ mice after MCAO contributes to the protections, others and we previously showed that inhibiting ERK activation with MEK1 inhibitors prevented IEG response, epileptogenesis and stroke-associated progressive brain damage[46, 49, 50]. Taken together, Ras/ERK signaling is a major pathway in mediating excitotoxic brain damage, and is regulated down-stream of NMDARs by tau.

The absence of Ras-GTP after stimulation of tau$^{-/-}$ neurons indicated a role of tau in excitotoxic NMDAR signaling upstream of Ras. SynGAP1 is found exclusively at post-synaptic sites in neurons, where it interacts with PSD-95 and inhibits Ras

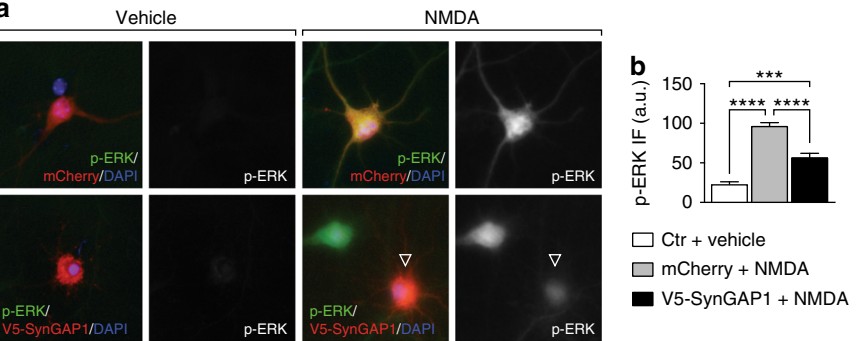

**Fig. 8** Expression of SynGAP1 mitigated NMDA-induced ERK activation in primary neurons. **a** Primary neurons transiently over-expressing V5-SynGAP1 (*red*) showed less ERK phosphorylation (p-ERK, *green*) in response to NMDA exposure, compared to control cells expressing mCherry (*red*) (or untransfected cells). No p-ERK was detected prior to NMDA treatments. **b** Quantification of fluorescence intensity of p-ERK signals in transfected neurons showed a significant reduction of ERK activation after NMDA in V5-SynGAP1− compared to mCherry-expressing neurons. Controls (ctr) comprised both mCherry, SynGAP1 or untransfected cells (***$p < 0.001$; ****$p < 0.0001$; $N = 3$; one-way ANOVA (Tukey post hoc)). All *error bars* are s.e.m

activity[42, 43]. The latter function of SynGAP1 is modulated by CaMKII-mediated and cdk5-mediated phosphorylation, which drives SynGAP1 out of post-synaptic densities, thereby possibly preventing excessive NMDAR-mediated Ras activation[43, 51–54] and regulating AMPA receptor trafficking[55]. SynGAP1 belongs to the Ras GTPase activating protein (RasGAP) family, which inactivate Ras by catalyzing GTP hydrolysis, bringing Ras back into its inactive, GDP-bound state[44]. Surprisingly, but in line with absent Ras activation in tau[−/−] neurons, we found markedly elevated levels of SynGAP1 at the post-synapse in complexes with PSD-95 in tau[−/−] brains. Transient cerebral ischemia in rats resulted in persisting (up to 24 h) phosphorylation of SynGAP1[56, 57], promoting increased interaction with Fyn paralleled by decreased PSD-95/SynGAP1 interaction[56]. This may reduce SynGAP1-mediated inhibition of excitotoxic Ras activation and therefore contribute to neuronal damage[56]. We have previously shown that synaptic Fyn levels are markedly reduced in tau[−/−] mice[21], possibly exacerbating the Ras-inhibitory effect of increased SynGAP1 levels during transient MCAO. Interestingly, pathogenic mutations in *SYNGAP1* have been linked to mental retardation[58] and distinct forms of epilepsy[59, 60]. Both, pathogenic missense and truncation mutants of *SYNGAP1* failed to reduce activity-dependent ERK phosphorylation when expressed in neurons, suggesting a loss-of-function mechanism[60]. Similarly, ERK activation is increased in primary neurons obtained from SynGAP1[−/−] mice[61] and by siRNA-mediated knockdown of SynGAP1 in primary wild-type neurons after NMDAR excitation[62]. While homozygote SynGAP1-deficient mice die postnatally[63], heterozygote SynGAP1-deficient mice are characterized by cognitive defects and spine maturation defects[64–67]. Importantly, we showed that reducing SynGAP1 levels in tau[−/−] mice by AAV-mediated shRNA expression, reinstated their susceptibility to induced seizures and progressive neurological deficits with brain damage following transient MCAO. The degree of seizures, neurological deficits and brain damage in AAV-SG1-shR-injected tau[−/−] mice following PTZ administration and MCAO were nearly as prominent as in tau[+/+] mice. This suggests that increased SynGAP1 levels at the post-synapse of tau[−/−] mice contribute significantly to their protection threshold, likely by providing a constant and strong inhibition of Ras activation and therefore disrupting Ras/ERK pathway in excitotoxic NMDAR signaling.

A number of studies reported changes in tau phosphorylation after ischemia, focusing on long-term effects of pathologically phosphorylated tau (AD-like mechanisms) on memory[14–19]. Consistent with these prior studies, we showed increased tau phosphorylation upon PTZ-induced excitotoxicity and MCAO in wild-type mice (Supplementary Fig. 9). However, our findings that reducing SynGAP1 levels and reinstating an excitotoxic ERK response in tau[−/−] mice increased susceptibility to induced seizures and abolished the protection from MCAO-mediated brain damage despite the absence of tau, supports that tau phosphorylation does not contribute significantly to the acute excitotoxic deficits. A mechanistic role of tau phosphorylation in particular in the long-term deficits following stroke remains to be shown.

In summary, we showed that tau[−/−] mice were protected from reperfusion damage induced by transient MCAO. In our study, we focused on tau-dependent excitotoxicity and found reduced excitotoxic Ras/ERK activation down-stream of NMDARs together with a concomitant increase of Ras-inhibiting SynGAP1 at the post-synapse in tau[−/−] mice, suggesting compartment-specific Ras inhibition. This suggests that tau controls post-synaptic SynGAP1 and therefore NMDAR-dependent Ras/ERK signaling in neurons. Our findings of virtually complete protection of tau[−/−] mice from acute brain damage in stroke with reperfusion introduce a new role for tau in this context and indicate a critical physiological role for control of post-synaptic compartmentalization of SynGAP1. Finally, our data suggest tau and SynGAP1 as potential drug targets in acute brain damage from stroke, therefore making targeting tau-dependent processes relevant beyond progressive age-related neurodegenerative disorders, such as AD.

## Methods

**Mice**. Tau[−/−] mice were generated by knockin of green fluorescence protein (GFP) encoding cDNA into the first exon of the endogenous *Mapt* locus as described before[68] (available from JaxMice #004779). Mice were maintained on a C57Bl/6 background. Three to 6 months old male mice were used throughout the study at indicated N-numbers. Experimenters were blinded to the randomly assigned genotype or type of AAV injected for all experiments until after analysis was completed. Blinding and sample/animal randomization was done by staff not involved in the study. All procedures were approved by the Animal Ethics Committee of the University of Sydney and the University of New South Wales, Australia.

**Middle cerebral artery occlusion**. Transient MCAO was used to induce stroke in adult mice[69]. Accordingly, male C57Bl/6 mice (age: 3–6 months; body weight: 25–35 g) were anesthetized and placed on their backs to expose the neck area. The common (CCA), external (ECA) and internal carotid arteries (ICA) were exposed via a ventral midline neck incision and connecting tissue around the vasculature was removed. The distal ECA was tied off and opened by arteriotomy and a heat-blunted 5-0 nylon monofilament was inserted and gently advanced upwards, ~10 mm past the CCA bifurcation. Reduction in cerebral blood flow, as determined by transcranial laser Doppler flowmetry (Moor instruments), confirmed MCAO. The

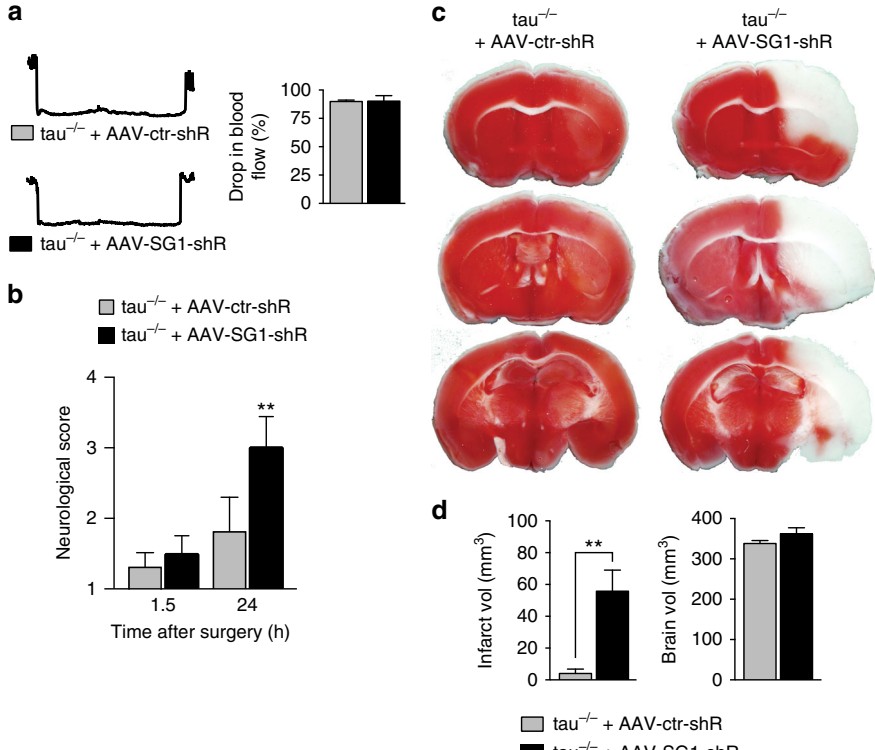

**Fig. 9** Reducing SynGAP1 levels in tau$^{-/-}$ mice confers susceptibility to transient MCAO. **a** Blood flow was reduced to the same extend during transient MCAO in 2 month-old tau$^{-/-}$ injected with either AAV-SG1-shR or AAV-ctr-shR at P1-3 (not significant; $N = 5$; Student's $t$-test). **b** Neurological scoring (with higher numbers indicating more severe impairments) revealed similar deficits directly after transient MCAO in AAV-SG1-shR- and AAV-ctr-shR-injected tau$^{-/-}$ mice, but thereafter, neurological deficits in AAV-SG1-shR-injected tau$^{-/-}$ mice progressed more severely than in AAV-ctr-shR-injected tau$^{-/-}$ mice (**$p < 0.01$; $N = 5$; two-way ANOVA (Tukey post hoc)). **c** Representative TTC-stained brain sections of tau$^{-/-}$ mice injected with AAV-SG1-shR or AAV-ctr-shR 24 h after transient MCAO (viable tissue stains *red*). Note the large infarcted area (*white*) in AAV-SG1-shR-injected tau$^{-/-}$ mice, while only minimal brain damage was present in AAV-ctr-shR-injected tau$^{-/-}$ mice. **d** Infarct and brain volume quantification of serial TTC-stained sections from AAV-SG1-shR-injected compared to AAV-ctr-shR-injected tau$^{-/-}$ mice (**$p < 0.01$; $N = 5$; Student's $t$-test). All *error bars* are s.e.m

monofilament was withdrawn after 1.5 h or 30 min (for long-term follow up) under continued anesthesia. Body temperature was maintained and monitored by placing mice on rectal probe-controlled heat pads (Kent Scientific Corporation) for the duration of the entire procedures. Mice were individually housed after full recovery from anesthesia. Neurological severity scoring (NSS) was done at indicated time points, according to the following (Supplementary Fig. 1): grade 0, no deficits; 1, decreased resistance to lateral push; 2, limb extension; 3, limb elevation; 4, circling. Animals were terminated at indicated time points after onset of MCAO. Mice for long-term follow up (30 min MCAO) were weighed and neurologically scored daily. In sham operated controls, the carotid arteries where exposed and the ECA cauterized before the skin was closed again. Mice that had bleeding complications during the surgery or when the filament was removed were excluded from the study and not counted to the total numbers examined. This occurred in less than 10% of mice, and as frequently in tau$^{-/-}$ and tau$^{+/+}$ mice. Blood pressure was recorded from the tail with CODA Surgical Monitor. Heart rate, peripheral $O_2$ saturation and body temperature were monitored using a PhysioSuite system (both Kent Scientific Corporation). Blood gases and electrolytes from whole blood were analyzed with an i-STAT Handheld device and CG8 + cartridges (Abbott).

**Rota-rod testing**. Rota-Rod testing of mice that have undergone 30 min of transient MCAO was performed as previously described[70]. Mice were trained on the accelerating-mode Rota-Rod (4–40 r.p.m. over 4 min) for 5 consecutive days prior to the surgery. Testing was done every 2 days after MCAO.

**Cortical NMDA infusion**. Stereotaxic surgeries were done as previously described[71]. Briefly, mice were anaesthetized with isoflurane and mounted in a stereotaxic frame (KOPF). The skin over the skull was opened and the bone exposed and cleaned. A small burr hole to allow injection into the brain was drilled into the skull. Coordinates for cortical injections were: AP: −2 mm, RL: 1.2 mm, DV: 1 mm. NMDA (50 mM; 0.2 µl) was infused over 1 minute into the cortex of mice under isoflurane anesthesia. The needle was left in place for another 5 min. After the surgery, mice were removed from the frame and the skin closed with sutures. All surgeries were done under aseptic conditions. Mice were transcardially perfused 24 h after the surgery, brains removed, paraffin embedded and serial

sections stained with a standard Nissl protocol. The area damaged was determined on 10 serial 10 µm sections at 100 µm intervals.

**Adeno-associated virus vectors**. A KpnI-linkered DNA fragment, entailing the mouse U6 promoter and a SynGAP1 small hairpin (sh) RNA (ccagaaccctctcttccatat), was synthesized (Epochbiolabs, Missouri City, USA) and cloned into a rAAV plasmid containing the CAG promoter driving a humanized renilla GFP reporter (Adeno-associated virus (AAV)-SG1-shR). The same backbone with an EGFP shRNA replacing SynGAP1 served as a control (AAV-ctr-shR). Packaging of rAAV1 vectors was performed as described[72]. One µl of either AAV-SG1-shR or AAV-ctr-shR vector ($2 \times 10^{12}$ viral genomes/ml) was injected bilaterally into the striatum ( + 4.0mm AP, $\pm$ 1.8 mm ML, −2.3 mm DV from lambda), thalamus ( + 2.0 mm AP, $\pm$ 1.7 mm ML, −2.5 mm DV) and cerebellum (−2.3 mm AP, $\pm$ 2.0 mm ML, −2.8 mm DV) of cryo-anaesthetized neonatal mice as described[73].

**Electroencephalography**. Methodology of electroencephalography has been previously described[74]. Briefly, after anesthesia with ketamine/xylazine and induction of MCAO, the recording electrode on remote telemetric transmitters (DSI) was implanted in the cornu ammonis (CA) region of the hippocampus (−2.0 mm AP, + 2.0 mm ML, −2.0 mm DV from bregma) and the reference electrode placed above the cerebellum (−6.0 mm AP, 0 mm ML, 0 mm DV). Local field potentials (LFPs) were recorded through amplifier matrices (DSI) at 500 Hz sampling rate (Dataquest A.R.T.). Raw LFPs were noise filtered using a powerline noise filter (DSI). Epileptiform discharge analysis of EEG recordings was performed with NeuroScore software v3.0 (DSI) with integrated spike detection module. Fast Fourier transform-based spectral analysis of interictal sequences was performed using NeuroScore software v3.0 (DSI). Average amplitude envelope time series were obtained by Hilbert transformation of filtered LFPs (MATLAB).

**PTZ administration**. To induce excitotoxicity, 6 weeks-old mice were administered pentylenetetrazole (PTZ; 50 mg/kg bodyweight i.p.)[21]. Directly after the injection, mice were individually placed into a 40 × 40 cm box to observe the development of seizures. Seizure severity rating was undertaken by an independent, blinded person

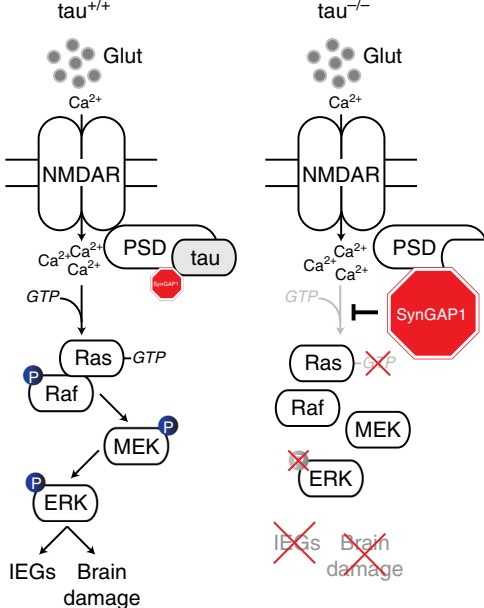

**Fig. 10** Schematic of excitotoxic, NMDAR-mediated Ras/ERK signaling in tau$^{+/+}$ and tau$^{-/-}$ mice. In wild-type mice (tau$^{+/+}$), excessive synaptic glutamate (glut) levels trigger increased calcium (Ca$^{2+}$) influx via NMDARs at the post-synapse. This triggers Ras activation by GTP binding, and eventually activation (phosphorylation; P) of ERK via the Ras/Raf/MEK/ERK cascade. To allow for Ras activation, levels of the site-specific inhibitor SynGAP1 (*red*) that resides in a complex with PSD-95 and tau need to be controlled (a process we found to be dependent directly or indirectly on tau). In contrast, our data revealed absence of excitotoxic Ras activation and hence downstream ERK phosphorylation in tau$^{-/-}$ mice. In the absence of tau, SynGAP1 is increasingly bound to PSD-95 and completely suppresses site-specific NMDAR-mediate activation of Ras at the post-synapse. As a consequence, neuronal damage and activation of immediate-early genes (IEGs) after stroke and excitotoxic seizures are suppressed

as follows: 0, no seizures; 1, immobility; 2, tail extension; 3, forelimb clonus; 4, generalized clonus; 5, bouncing seizures; 6, full body extension; 7, status epilepticus.

**Primary neuronal cultures and staining**. Primary neurons were obtained from 16 days-old tau$^{+/+}$ and tau$^{-/-}$ embryos[75]. Briefly, the abdominal cavity of time-mated females was opened to remove the uterus. Embryos were placed on ice, decapitated and brains removed. After meninges were carefully removed, cortices and hippocampi were dissected and incubated with trypsin (Sigma) at 37 °C for 15–20 min, followed by trituration with fire-polished glass Pasteur pipettes (Livingstone) to obtain single cell solutions. Cells were counted using a hemo-cytometer and plated in Dulbecco's Modified Eagle Medium (Life technologies) medium containing 10% heat-inactivated fetal bovine serum (Hyclone). Medium was changed to Neurobasal containing B27 supplement and Glutamax (all Life technologies) for continued culturing. Neurons were cultured for 15 days, and then treated with either 10 μM NMDA, 25 μM NMDA, 50 mM KCl, 5 μM forskolin or 50 μM bicuculline for 30 min before harvesting for Western blotting. To determine cell death, cells were treated with 0, 10 or 25 μM NMDA for 30 min. Then, NMDA-containing medium was removed and cells were washed twice with warm Neurobasal medium (Thermo) before conditioned medium from before treatments was added back. After a further 24 h incubation, cells were fixed with 4% PFA and mounted in Fluoromount-G (SouthernBiotech) with DAPI (Molecular Probes). Cells with condensed nuclei were considered dead[76]. For staining, cells were fixed at 21 days in vitro (DIV) with 4% PFA and stained with primary antibodies to SynGAP1 (Sigma) and PSD-95 (Millipore) using established protocols[77]. Images were taken with an Eclipse Ti confocal microscope (Nikon). DIV 4 primary neurons were transfected V5 tagged SynGAP1 (V5-SynGAP1) or mCherry control pLVX expression plasmids using Lipofectamine LTX (Invitrogen) according to the manufacturers protocol. DIV 12 neurons were pre-treated with 5 μM nifedipine, 40 μM CNQX and 1 μM tetrodotoxin citrate (all from Tocris) for 1 h, followed by treatment with 100 μM NMDA (or vehicle) for 6 min at 37 °C/5% CO$_2$. Primary antibodies for staining were against phosphorylated ERK (Cell Signaling) and V5 (Sigma), and DAPI was used for nuclear visualization. Phosphorylated ERK staining intensity of randomly selected transfected neurons was quantified using

ImageJ (NIH). Data from vehicle treated cells were pooled, since there was no difference between transfections.

**Calcium imaging**. Mice were injected at P0 with AAVs expressing the Ca$^{2+}$ reporter GCaMP5G. At 1 month of age, acute brain slices (400 μm) were prepared using VT1200 vibratome (Leica) according to standard procedures. Briefly, mice were sacrificed, brains removed and sectioned coronally in modified high sucrose low sodium ice cold artificial cerebrospinal fluid (sACSF) containing 4 mM KCl, 1 mM CaCl$_2$, 6 mM MgCl$_2$, 25 mM NaHCO$_3$, 246 mM sucrose, 10 mM glucose and the pH indicator phenol red (pH adjusted to 7.3), bubbled with carbogen (95% O$_2$, 5% CO$_2$). Slices were thereafter maintained at room temperature in artificial cerebral spinal fluid (ACSF) solution containing 119 mM NaCl, 2.5 mM KCl, 2.5 mM CaCl$_2$, 1.5 mM MgCl$_2$, 26 mM NaHCO$_3$, 1 mM NaH$_2$PO$_4$, and 11 mM glucose, bubbled with carbogen. After equilibration of at least 60 min, slices were transferred onto a recording chamber and constantly superfused at 2 ml/min ACSF bubbled with carbogen. After recording baseline responses for 5 min, slices were exposed to 1 mM glutamate (bath applied in ACSF) for 5 min and cortical Ca$^{2+}$ responses were monitored until neurons recovered back to baseline levels. Changes in GCaMP5G fluorescence in the cortex were imaged using a confocal microscope (Zeiss 710NLO LSM, 488 nm excitation; 5x/0.3 W objective). Images were taken every 10 seconds. The Zen software (Zeiss) was used to measure mean pixel intensity of the whole field.

**Histology**. Immunohistochemical staining and quantification of fluorescence intensity has been previously described in detail[78]. Briefly, paraformaldehyde fixed and paraffin embedded tissue was section on a microtome (Thermo) to 5 μm. Sections were rehydrated via xylene followed by decreasing concentrations of ethanol. For staining, sections were individually mounted in Sequenza racks (Thermo), blocked with 2% heat-inactivated goat serum (Sigma)/3% bovine serum albumin (Sigma) in PBS, before incubation with primary antibodies. Primary antibodies were visualized by incubation with Alexa-fluorophore labeled secondary antibodies (1:250, Molecular Probes) after washing with PBS. Primary antibodies were against SynGAP1 (1:100, Sigma), MAP2 (1:500, Sigma), Arc (1:100, Santa-Cruz), pH2AX (1:200, Chemicon), NeuN (1:500, Chemicon), hrGFP (1:250, abcam) and Tau5 (1:250, Invitrogen). DAPI (Molecular Probes) was used for nuclear counterstaining. 1mm fresh brain slices were obtained with a brain blocker (KOPF) and stained for 10 min at 37 °C with a 2% TTC/PBS (Sigma) solution until viable tissue turned bright red. Fluorescence intensity and infarct size were determined using ImageJ (NIH). Infarct sizes were adjusted for cerebral edema using the contralateral hemisphere as control.

**Cerebral vasculature visualization**. Cerebral vasculature staining was performed using Indian ink gelatin solution[79]. Briefly, deeply anesthetized mice were perfused with PBS and cold 4% PFA via left ventricular puncture followed by slow infusion of 0.5–1 ml 50% Indian ink in 5% gelatin at a rate of 1 ml per 30 s. Perfusion was stopped prior to ink returning to the right atrium to reduce cerebral venous filling. Mice were then left to chill on ice for 10 min to allow the gelatin to set prior to careful removal of the brain.

**Western blotting**. For Western blotting, protein extracts were separated by SDS-PAGE followed by semi-dry transfer onto 0.2 μm nitrocellulose membranes (Invitrogen)[77]. Membranes were blocked with 5% bovine serum albumin (Sigma) in TBS, washed with TBS containing 1% Tween-80 (Sigma) and then incubated with primary antibodies in blocking buffer. Primary antibodies were against ERK (1:1000, Sigma), SynGAP1 (1:1000, Sigma), phospho-ERK (1:500, Cell Signaling), V5 (1:5000, Invitrogen), Tau5 (1:1000, Invitrogen), pS214 (1:1000, Invitrogen), pS422 (1:2000, Invitrogen), pS396/pS404 (1:1000, PHF-1, P. Davies), RAS (1:1000, Millipore), Psd95 (1:2000, Millipore) and Gapdh (1:5000, Millipore). Blots were visualized by HRP-coupled secondary antibodies (1:5000, Sigma), with Luminata Crescendo Western HRP substrate (Millipore), and detected and quantified in a VersaDoc Model 4000 CCD camera (BioRad) or a ChemiDoc MP system (BioRad). Membranes were stripped for re-probing as previously described[77]. Full membranes of all Western blots presented are provided in the Supplementary Fig. 10.

**RNA purification and quantitative PCR**. A RNeasy Mini Kit (Qiagen) was used to extract total RNA from mouse brain tissue and primary cultured neurons, following the manufacturer's instructions. To remove contaminating genomic DNA, an on-column DNA-digest was performed with RNase-free DNase I (Qiagen). cDNA was synthesized from 2.5 μg of total RNA with the second strand cDNA-synthesis kit (Invitrogen). mRNA levels were determined by quantitative PCR, using a Fast SYBR green reaction mix (Invitrogen) and gene-specific primer pairs as listed in Supplementary Table 3, using a Mx3000 real-time PCR cycler (Stratagene).

**Transcriptome and pathway analysis**. Next generation RNA sequencing (RNA-Seq) was done by BGI-Hong Kong (China) using RNA extracted from vehicle-injected tau$^{+/+}$ and tau$^{-/-}$ and PTZ-injected tau$^{+/+}$ and tau$^{-/-}$ mice. PTZ mice with similar seizure score were selected for this analysis. Briefly, at least

24 million, 90 bp long read pairs per sample could be aligned unambiguously to the GRCm38/mm10 version of the mouse genome using tophat 2.03[80] and bowtie 2.0.0.6[81] and allowing for 2 read mis-matches. Differential expression analysis was performed using Cuffdiff 2.01[80] and only genes with a $p$-value of less than 0.05 and a fold change of more than 1.5-fold were labeled as significantly differentially expressed. Genes were labeled as lack of response, if genes were significantly differentially expressed in vehicle-injected compared to PTZ-injected tau$^{+/+}$ mice and lacking or having a significantly milder response in the same direction in vehicle-injected compared to PTZ-injected tau$^{-/-}$ mice. Functional annotation of the significant RNA-Seq genes was performed using DAVID[82]. KEGG pathway[83] representations were used to represent the outcome of the analysis. All sequencing data have been submitted to the GEO repository and are available under accession number GSE45703.

**Active ras pull-down.** GTP-bound Ras was precipitated from stimulated hippocampal slices as previously described[84]. Briefly, 2-month-old mice were sacrificed, brains removed and transferred into $CO_2$-adjusted and ice-cold sucrose cutting solution (0.2 mM $CaCl_2$, 7 mM $MgCl_2$, 28 mM $NaHCO_3$, 11 mM glucose, 1.25 mM $NaH_2PO_4$, 2.5 mM KCl and 241 mM sucrose). The hippocampi were removed, sliced and then incubated in $CO_2$-adjusted artificial cerebral spinal fluid (aCSF) (127 mM NaCl, 2.5 mM KCl, 1.25 mM $NaH_2PO_4$, 1 mM $MgCl_2$, 25 mM $NaHCO_3$ and 25 mM glucose) for 30 min at 37 °C. Slices were stimulated at room temperature for 10 min in modified aCSF containing 62.5 mM KCl, 4 mM $CaCl_2$, no $MgCl_2$, 10 μM CNQX, 5 μM d-AP, and 1 μM TTX. Reactions were terminated by replacing the medium with ice-cold lysis buffer (25 mM HEPES (pH 7.5), 150 mM NaCl, 1 % nonidet P-40, 0.25 % Na$^+$-deoxycholate, 10 % glycerol, 10 mM $MgCl_2$, 1 mM EDTA and protease inhibitor (Roche)). Lysates (100 μg) were incubated with recombinant Raf-RBD coupled to beads to precipitate activate Ras.

**Co-immunoprecipitation.** Interaction of proteins was determined by co-immunoprecipitation experiments[21]. Briefly, brain tissue or cells were homogenized in a buffer containing 50 mM Tris-HCl, 150 mM NaCl, 1% NP-40 (all Sigma) and complete proteinase inhibitor (Roche). After clearing by centrifugation, 200 μg of protein was incubated with antibodies over night at 4 °C. Antibodies used for precipitation were against SynGAP1 (1:200, Sigma), Tau1 (1:200, Millipore), Tau5 (1:200, Invitrogen), RasGRF (1:200, SantaCruz) and 4RTau (1:200, Dako). Antibodies were then captured with magnetic protein G beats (Invitrogen) and washed with lysis buffer and increasing NaCl concentrations (150-250-450 mM) before adding sample buffer for subsequent Western blotting. HEK293T cells (ATCC) were transiently transfected with Flag-PSD-95, tau and V5-SynGAP1 expression plasmids as previously described[22].

**Synaptosome preparations.** Synaptosomes were purified from mouse brains using a differential extraction procedure[21]; First, tissue was homogenized on ice in a Sucrose Buffer containing 0.32 M sucrose, 1 mM $NaHCO_3$, 1 mM $MgCl_2$ and 0.5 mM $CaCl_2$. Then, homogenates were cleared by two rounds of centrifugation (1400 × $g$/10 min/4 °C). The supernatants from both spins were combined, cleared again by centrifugation (720 × $g$/10 min/4 °C), and then crude synaptosomes were pelleted by high-speed centrifugation (13,800 × $g$/10 min/4 °C). Pellets were resuspended in 300 μl Sucrose Buffer, layered on top of 1 ml pre-cooled 5% Ficoll and high-speed centrifuged (45,000 × $g$/45 min/4 °C). Supernatant were discarded, pellets resuspended in 100 μl pre-cooled 5% Ficoll, and layered on top of 1 ml pre-cooled 13% Ficoll for the next high-speed centrifugation (45,000 × $g$/45 min/4 °C). The resulting interface contained the purified synaptosomes, and was recovered carefully. Purified synaptosomes were extracted from the interfaces by diluting them with Sucrose Buffer followed by pelleting with high-speed centrifugation (45,000xg/45 min/4 C). Pure synaptosomes were further fractionated to obtain soluble, membranous and PSD-associated proteins; Therefore, pellets were resuspended in 40 mM Tris-HCl (pH 6) containing 2% Triton X-100, 0.5 mM $CaCl_2$ (all Sigma) and complete protease inhibitors (Roche), followed by incubation (15 min/4 °C) and high-speed centrifugation (40,000 × $g$/30 min/4 °C). The supernatants were recovered as soluble protein fraction. Pellets were washed, using the same 40 mM Tris-HCl buffer, incubation and centrifugation conditions as in the prior step. Then, the pellets were resuspended in 20 mM Tris-HCl (pH 8) containing 100 mM NaCl, 1 mM EGTA, 1 mM EDTA, 0.5% sodiumdeoxycholate, 0.1% SDS, 1% Triton X-100 (all Sigma) and complete protease inhibitors, followed by incubation (15 min/4 °C) and high-speed centrifugation (40,000 × $g$/30 min/4 °C). The supernatants were recovered as membranous protein fraction. Again, pellets were washed using the conditions of the prior step. The final extraction was done by resuspending the pellets in 5% SDS, sonication and high-speed centrifugation (20,000 × $g$/10 min/4 °C). The final supernatants resembled the PSD-associated protein fraction.

**Duolink proximity ligation assay.** Primary neurons were fixed at 21 DIV with 4% PFA for 20 min and permeabilized with 0.1% Triton X-100 in PBS for 5 min. Cells were incubated in Duolink blocking solution (Olink Bioscience) for 30 min at 37 °C, followed by incubation with monoclonal mouse Tau5 (Invitrogen), rabbit SynGAP1 (Sigma) and chicken β3-tubulin (Chemicon) primary antibodies diluted in Duolink Antibody Diluent at for 1 h room temperature. The ligation assay was then conducted according to the manufacturer's instructions (Olink Bioscience)

and β3-tubulin was detected using A488-labeled anti-chicken secondary antibody (Molecular Probes). Images were taken with an Eclipse Ti confocal system (Nikon).

**Statistics.** Pre-study sample size calculation was based on decreased susceptibility of tau$^{-/-}$ mice to induced excitotoxic seizures, previously shown by us[21]. To detect a 40% reduction in infarct size ($\sigma = 0.2$) with a power of 0.95 and $\alpha = 0.05$ we required a sample size of 8 ($N = 7.61$, Cohen method). Based on our tau$^{-/-}$ MCAO data, we calculated a pre-study sample size of 4 ($N = 3.21$) to detect a 75% difference in infarct size for the SynGAP1-knockdown MCAO study. Statistical analysis of results was done with the Prism 6 software (GraphPad Software, USA), with tests used indicated in figure legends. Values are given as mean ± s.e. All experiments were repeated at least three times.

**Data availability.** All sequencing data have been submitted to the GEO repository and are available under accession number GSE45703. All other relevant data are available from the authors upon request.

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

## Acknowledgements

The authors thank Dr Vladimir Sytnyk for help with confocal microscopy, the staff of the Biological Resources Center Wallace Wurth animal facility for continuing support with mice, Dr Peter Davies for antibodies, and Dr David W Howells, Dr Geoffrey A Donnan (both: Florey Institute of Neuroscience and Mental Health, University of Melbourne, Australia), Dr Edna Hardeman (University of New South Wales, Australia) and Dr Nikolas Haass (University of Queensland, Australia) for helpful comments on the manuscript. This work was supported by funding from the National Health and Medical Research Council (NH&MRC), the Australian Research Council (ARC), the Alzheimer Association (US), Alzheimer's Australia, the Jane Mason & Harold Stannett Williams Memorial Foundation (Australia) and the University of New South Wales. L.M.I. is an NH&MRC Senior Research Fellow. Y.D.K. is an NH&MRC Career Development Fellow. J.v.E. is an ARC DECRA fellow.

## Author contributions

L.M.I. and Y.D.K. designed and supervised the project. M.B., A.G., J.v.E., A.I., M.P., A.v.H., S.W.C., J.v.d.H., W.S.L., J.P., T.F., G.v.J., H.S. and Y.D.K. performed experiments. J.M. analyzed the sequencing data. E.G., G.D.H. and M.K. provided materials and expertise. L.M.I., M.B. and Y.D.K. wrote the manuscript. A.G., J.v.E., T.F., J.M., E.G., G.D.H. and M.K. helped with the editing of the manuscript.

## Additional information

**Competing interests:** The authors declare no competing financial interests.

