## [Peer review file · Nature Communications]

Reviewers' comments:

Reviewer #1 (Remarks to the Author):

In this study Gladbach et al. investigate the role of tau in two brain injury models with an excitotoxic component, namely stroke and seizures. In a transient MCA occlusion model, tau-/- mice showed remarkable protection and behavioral improvement at 24 hrs after reperfusion. Similarly, PTZ induced seizure activity was significantly reduced in tau-/- mice. The identification of the Ras/ERK signaling pathway by transcriptomic profiling of PTZ injected mice is very logical and well performed. The authors convincingly show that the GTPase-activating protein SynGAP1 directly or indirectly interacts with tau and that PSD95-associated SynGAP1 is increased in tau-/- mice resulting in reduced Ras/ERK activation and excitotoxicity. The reversal of the protection observed after shRNA delivery confirms the mechanistic importance of this pathway. Overall the data are of good quality and are supportive of the main hypothesis. The study clearly shows tau involvement in mouse seizure and stroke models. On the other hand, as mentioned below, there are better ways to test NMDAR excitotoxicity. The concerns are as follows:

1. The role of NMDA receptors in the injury models in vivo and in vitro should be addressed more directly. In addition to the stroke and seizure model, one could show how the brain of tau-/- mice reacts to cortical NMDA injections. At the same time, NMDA can be used as an excitotoxic agonist in vitro too. This would more concisely associate tau to the NMDA receptor and not to other excitotoxic glutamate receptors. GABA antagonism is not a direct excitotoxic stimulus and unnecessarily complicates the main message that tau is a PSD95 organizer for Ras signaling.
2. Regarding NMDA-tau receptor coupling under excitotoxic conditions. The authors need to show that ionflux (Ca²⁺) through the receptor is not affected by tau deletion. This is to assure that downstream signaling cascades are equally activated in WT and tau-/- mice.
3. The ERK pathway has generally been implicated in pro-survival signals associated with NMDR activation. What are the cell death effectors downstream of ERK?
4. The authors do not show LDF blood flow traces during the reperfusion phase. How do WT and tau-/- mice compare during this critical period? From the trace shown in figure 1 one can see that the tau-/- trace seems to dip at the end of the recording. Alteration in neuronal activity might have implications for post-ischemic blood flow regulation, which could impact stroke outcome. Reperfusion blood flow is a critical factor in the tMCA occlusion model and should be monitored for 30 minutes.
5. In supplemental figure S1 authors show the loss of neuronal Map2 signal and increased glial activation in the hippocampus of WT mice 3 hours after MCAO occlusion. This observation is somewhat puzzling because blood to the hippocampus is not supplied by the MCA and hippocampal cell death is not a consistent feature in this ischemia model. It does not make a strong case that this loss of MAP2 reactivity is due to excitotoxic cell death. The authors would have to show that similar is observed in the ischemic territory.
6. The immunofluorescent micrographs in figure 4B are of insufficient quality. It seems there is a problem with the tissue quality. SynGAP1 staining has high background levels. In addition, whether GFP expression is predominantly in neurons is not evident. Barographs are missing.
7. Comparing TTC stained brain sections at day 1 and 5, it seems that the infarct grows somewhat. Are there any quantitative data for that and what are the implications?
8. Are there any previous studies that investigated the cerebral vascular anatomy in tau-/- mice. If not, the authors are encouraged to present basal and dorsal aspects of the brain vasculature of WT and tau-/- mice as a supplementary figure.

Reviewer #2 (Remarks to the Author):

This is a comprehensive assessment of the effects of tau knockout on brain damage from transient middle cerebral artery occlusion (tMCAO) in mice and an assessment of the role of Ras-inhibiting SynGAP1 in the protective effects of tau deletion. Most of the studies appear to be well powered, with robust effects and consistency among the data sets of similar endpoints as well as potential underlying mechanism(s). There are a few issues that the authors need to address which are categorized as major issue and minor issues below.

Major Issues

1. Statistical analysis: The use of student t tests for analysis of complex experimental design is inappropriate. For the described design and results reported, the student t test is an inappropriate statistical approach with which to evaluate group differences for the current series of experiments. Specifically, in the methods section, the authors describe designs involving the assessment of two genotypes (tau+/+ or tau -/-), two surgical conditions (Sham or MCAO) and/or two AAV vector conditions (SG1-shR or ctr-shR) on dependent variables of interest, in several instances being assessed at multiple time points following when experimental manipulations were induced. For example, in Figure 1E, the experimental design these data represent is that of a 2 way ANOVA with repeated measures (5 in total). In order to conduct assessments for group differences using t-tests as the author describes, they would have to have conducted 30 individual t-tests to compare each of the groups from each other at each time point. The issue with this is that the authors make no mention of any false discovery rate correction for the increased risk of making a Type I error (incorrectly rejecting the null hypothesis using sample data when there is not a difference in the population) that comes with multiple comparisons. Doing this number of t-test two-group comparisons for a given experiment would yield and 80% that at least one of them is a false positive (for example Fig 1E, but there are numerous other instances of this). The omnibus 2 way ANOVA with follow up contrasts based on the significance of main effects/interactions of these factors would protect against this issue and is the more appropriate approach here. The only instance in Figure 1 where uncorrected t-tests would be appropriate is in Figure 1B/I. This issue of the inappropriateness of the statistical test being applied is present throughout the results section/figures and as such, makes the interpretation of any of the data as they are presented currently a significant challenge and a major concern.

2. Tau phosphorylation state: It is not clear from the study description if the knockout of tau protects from stroke damage by removing total tau or prevention of the reported hyperphosphorylation of tau after stroke. This is a critical issue that needs to be addressed by the authors.

Minor Issues:

1. Figure 1B lacks a description (axis) of the units of cerebral blood flow.

2. It is not clear how animals were assigned into the various group after tMCAO. For lesion volume and neurological scores, 12 mice per group were used. For other measurements, much smaller group sizes were used (4-5). Despite these small group sizes, the variances were extremely small of nearly all outcome measures. As such, a description of the assignment of animals to these endpoint measures is needed.

3. The sentence on page 7, lines 7 to 8, does not make sense to this reviewer.

4. Page 8, last paragraph, lines 9-10, is an apparent dismissal of previous work on tau hyperphosphorylation following stroke. This sentence is not needed and should be removed.

Reviewer #3 (Remarks to the Author):

The study by Ittner and colleagues demonstrated that tau ablation protected mice from damages caused by MCAO, prevented upregulation of ERK pathways. They also provided evidence that the protection could be mediated by the enhanced SynGAP-PSD95 interaction, upregulated in TauKO mice. The protective effects are striking, and the involvement of SynGAP provided novel mechanistic insight. The overall data quality is excellent. The following concerns need to be addressed:

1. Only male mice are used. Is this phenomena only applicable to males? It would be nice to know.
2. Inhibition of SynGAP1 makes the tauKO mice vulnerable to MCAO. One concern is that SynGAP1 inhibition could be detrimental, thus the elevated toxicity could contribute to MCAO damage in tauKO mice via a parallel pathway. A complementary experiment could be more informative. Does overexpressing synGAP1 in tau+/+ mice render them protective against MCAO?

Minor comments:

1. The legends include non-necessary information (i.e discussion of the results etc.). Consider revise to be more concise.
2. The genes in Fig. 2C are not readable. Consider revising.

Reviewer 4 [Additional feedback; comments to the Authors]

The (stroke) data are extremely limited because of the lack of an extended time course and rudimentary outcomes (NSS only for behavior). It's unclear why both groups had such limited outcomes in terms of time course in all outcomes and behavior techniques. Long term outcomes and functional tests are the essential requirements for stroke research nowadays. And with the NSS data, it's surprising to see the score of 2 extending to 120h post-MCAO for the -/- mice. As this measures limb extension, this score is typically around zero by 3-5 days after MCAO with such small infarct. The study should show the 120 h timecourse for the +/+ mice. In addition, the entire Fig. 4 was done using the -/- mice. The study should include the wildtype tau mice for comparisons.

Point-to-point response to reviewer comments

We would like to thank the reviewers for their comment. We feel that addressing all of them has improved the manuscript, and we hope the reviewers share our view to recommend publication of our work. Please find a detailed response to each individual comment below.

Reviewer #1

In this study Gladbach et al. investigate the role of tau in two brain injury models with an excitotoxic component, namely stroke and seizures. In a transient MCA occlusion model, tau^{-/-} mice showed remarkable protection and behavioral improvement at 24 hrs after reperfusion. Similarly, PTZ induced seizure activity was significantly reduced in tau^{-/-} mice. The identification of the Ras/ERK signaling pathway by transcriptomic profiling of PTZ injected mice is very logical and well performed. The authors convincingly show that the GTPase-activating protein SynGAP1 directly or indirectly interacts with tau and that PSD95-associated SynGAP1 is increased in tau^{-/-} mice resulting in reduced Ras/ERK activation and excitotoxicity. The reversal of the protection observed after shRNA delivery confirms the mechanistic importance of this pathway. Overall the data are of good quality and are supportive of the main hypothesis. The study clearly shows tau involvement in mouse seizure and stroke models. On the other hand, as mentioned below, there are better ways to test NMDAR excitotoxicity. The concerns are as follows:

1. The role of NMDA receptors in the injury models *in vivo* and *in vitro* should be addressed more directly. In addition to the stroke and seizure model, one could show how the brain of tau^{-/-} mice reacts to cortical NMDA injections. At the same time, NMDA can be used as an excitotoxic agonist *in vitro* too. This would more concisely associate tau to the NMDA receptor and not to other excitotoxic glutamate receptors. GABA antagonism is not a direct excitotoxic stimulus and unnecessarily complicates the main message that tau is a PSD95 organizer for Ras signaling.

We appreciate the suggestion by the reviewer to use cortical NMDA injections as an additional experimental *in vivo* model to show the marked protective effects of tau reduction from excitotoxicity. As requested by the reviewer, we injected NMDA into the cortex of tau^{-/-} and tau^{+/+} mice. Consistent with smaller infarcts after MCAO, the brain damage 24 hours after unilateral stereotaxic injection of 0.2μl of 50μM NMDA into the cortex was significantly smaller in tau^{-/-} mice compared to tau^{+/+} animals. This new data has been included in the revised manuscript as new supplementary figure S6 and is discussed in the main text by stating “Focusing on excitotoxicity was further supported by larger brain damage after intra-cortical infusion of NMDA in tau^{+/+} compared to tau^{-/-} mice, ... (fig. S6).” (page 6, lines 35-37).

In addition to using NMDA for intra-cortical injections as outlined above, we have followed this reviewer’s suggestion to used NMDA for challenging primary neurons. Firstly, we determined the level of NMDA-induced toxicity by comparing number of dead cells 24 hours after the challenge between wild-type and tau^{-/-} primary cortical neurons. Neuronal death was assessed according to Zhou et al. (*Cell death & disease*, 2013) by counting numbers of intact versus pyknic nuclei stained with DAPI. Consistent with our *in vivo* findings, the number of dead neurons was significantly reduced in tau^{-/-} neurons compared to wild-type cells. This new data in primary neurons is presented in the new supplementary figure S6 and discussed in the main text by stating: “...and reduced NMDA-mediated neuronal death in tau^{-/-} compared with tau^{+/+} primary cultured neurons (fig. S6).” (page 6, lines 36-37).

Secondly, we determined the levels of ERK phosphorylation induced by NMDA in tau^{-/-} and wild-type neurons. Consistent with our previous findings with bicuculline, we found increased ERK phosphorylation in wild-type neurons challenged with 10 and 25 μM NMDA, while there was no corresponding increase in tau^{-/-} cells. We observed a decrease in ERK phosphorylation in NMDA treated tau^{-/-} neurons, which is consistent with a significant engagement of extra-synaptic NMDAR that inhibit ERK signaling [Ivanov A, *J Physiol* 2006], as compared to challenges with bicuculline that result in a predominantly synaptic activation of NMDAR (and hence no reduction in phospho-ERK in tau^{-/-} cells). Together, this provides further support for the concept that tau regulates synaptic NR2B-containing NMDARs [Ittner et al., *Cell* 2010; Ittner & Gotz, *Nature Rev Neurosci* 2011; Ittner et al., *Science* 2016]. This new data in primary neurons is presented in the revised Figure 4c,d and discussed in the main text by stating: “To directly assess NMDAR-mediated activation of ERK signaling, we treated primary neurons with NMDA (Fig. 4c,d). This resulted in increased levels of ERK phosphorylation in tau^{+/+} neurons, but no activation and rather decreased phosphorylation of ERK in tau^{-/-} cells. This is in line with a failure to activate ERK downstream of synaptic NMDARs, while ERK inhibition mediated by extra-synaptic NMDARs³⁶ remained intact in tau^{-/-} neurons.” (page 5, lines 22-25).

We feel that these new experimental paradigms have significantly strengthened our study.

2. Regarding NMDA-tau receptor coupling under excitotoxic conditions. The authors need to show that ionflux (Ca²⁺) through the receptor is not affected by tau deletion. This is to assure that downstream signaling cascades are equally activated in WT and tau^{-/-} mice.

As suggested by the reviewer, we provide new data on neuronal Ca^{2+} influx in the cortex in $\text{tau}^{-/-}$ and $\text{tau}^{+/+}$ mice upon NMDA-receptor activation. To measure neuronal Ca^{2+} flux in acute cortical slices from $\text{tau}^{-/-}$ and $\text{tau}^{+/+}$ mice, we expressed the Ca^{2+} reporter GCaMP5G in neurons using AAV-mediated gene transduction. Acute slices were challenged with 1mM glutamate. Consistent with unchanged sEPSC recording in $\text{tau}^{-/-}$ mice reported by us earlier [Ittner LM et al., *Cell* 2010], both $\text{tau}^{-/-}$ and $\text{tau}^{+/+}$ slices showed comparable profiles of initial Ca^{2+} responses after stimulation, supporting similar activation of NMDARs with differences rather in downstream signaling. This new data is presented as new supplementary figure S2 and discussed in the main text by stating “Confirming that neuronal excitability was not compromised in $\text{tau}^{-/-}$ brains *per se*, we measured comparable Ca^{2+} responses in brain slices of $\text{tau}^{+/+}$ and $\text{tau}^{-/-}$ mice expressing the GCaMP5G Ca^{2+} reporter in neurons challenged with 1mM glutamate (fig. S2)” (page 3, lines 32-34).

3. The ERK pathway has generally been implicated in pro-survival signals associated with NMDR activation. What are the cell death effectors downstream of ERK?

ERK signaling has diverse functions, including pro-survival signaling down-stream of NMDA receptor activation. However, ERK signaling can have pro-apoptotic functions [Lu Z and Xu S, *IUBMB Life*, 2008]. Accordingly, persistent ERK activation is associated with glutamate-induced toxicity in neurons, which is prevented by ERK inhibition [Stanciu M et al., *Journal of Biological Chemistry*, 2000; Gladbach A et al, *Journal of Neural Transmission*, 2014]. Furthermore, ERK inhibition significantly reduced ischemia-induced brain damage [Alessandrini A et al., *PNAS*, 1999; Namura S et al., *PNAS*, 2001; Gladbach A et al, *Journal of Neural Transmission*, 2014]. Therefore, prolonged ERK activation contributes to excitotoxic damage of neurons. How ERK activation promotes cell death remains largely unknown. It may involve caspase 3 activation, oxidative toxicity or mitochondrial cytochrome C release [Lu Z and Xu S, *IUBMB Life*, 2008]. To this end, we do not know the exact downstream effectors of ERK in the context of MCAO-induced brain damage, but our study – including the genomic data – may be a starting point for future studies to unravel the molecular cascades downstream of ERK. However, the present study focuses on tau-dependent events upstream of ERK, showing an unprecedented link via SynGAP1.

4. The authors do not show LDF blood flow traces during the reperfusion phase. How do WT and $\text{tau}^{-/-}$ mice compare during this critical period? From the trace shown in figure 1 one can see that the $\text{tau}^{-/-}$ trace seems to dip at the end of the recording. Alteration in neuronal activity might have implications for post-ischemic blood flow regulation, which could impact stroke outcome. Reperfusion blood flow is a critical factor in the tMCA occlusion model and should be monitored for 30 minutes.

We agree with the reviewer that similar reperfusion after transient MCAO is critical for comparing deficits in $\text{tau}^{-/-}$ and $\text{tau}^{+/+}$ mice. As requested by the reviewer, we extended the presentation of Doppler flow recordings to 30 minutes after the filament was removed. This showed comparable reperfusion in $\text{tau}^{-/-}$ and $\text{tau}^{+/+}$ mice. The extended traces are now presented in the revised Figure 1b.

5. In supplemental figure S1 authors show the loss of neuronal Map2 signal and increased glial activation in the hippocampus of WT mice 3 hours after MCAO occlusion. This observation is somewhat puzzling because blood to the hippocampus is not supplied by the MCA and hippocampal cell death is not a consistent feature in this ischemia model. It does not make a strong case that this loss of MAP2 reactivity is due to excitotoxic cell death. The authors would have to show that similar is observed in the ischemic territory.

As suggested by the reviewer, we replaced the original MAP2 staining in the hippocampus with MAP2 staining from the cortex of both wild-type and $\text{tau}^{-/-}$ mice. The loss of MAP2 staining 3 hours after transient MCAO observed in wild-type brains was prevented in $\text{tau}^{-/-}$ mice. This data is now presented in the revised supplementary figure S3 and referred to in the main text by stating: “Early loss of neuronal microtubule-associated protein 2 (MAP2) staining has been reported after stroke³². Accordingly, MAP2 staining was profoundly and broadly reduced in the ischemic cortex of $\text{tau}^{+/+}$ mice 3h after transient MCAO, while a moderate reduction of MAP2 staining in $\text{tau}^{-/-}$ brains was confined to a small core area without changes in areas of the cortex that corresponded to those affected in $\text{tau}^{+/+}$ mice (fig. S3).” (page 3/4, lines 37/1-4).

6. The immunofluorescent micrographs in figure 4B are of insufficient quality. It seems there is a problem with the tissue quality. SynGAP1 staining has high background levels. In addition, whether GFP expression is predominantly in neurons is not evident. Bar graphs are missing.

We improved the quality of the immunofluorescence staining presented in Figure 4B. Higher magnification showed GFP expression in neurons together with a marked reduction of SynGAP1 staining in the same areas, supporting successful knockdown of SynGAP1. Furthermore, higher magnifications show that SynGAP1 staining is mostly confined to synaptic structures (buttons), which appeared as ‘background’ on lower magnification images. We added scale bars as requested.

7. Comparing TTC stained brain sections at day 1 and 5, it seems that the infarct grows somewhat. Are there any quantitative data for that and what are the implications?

We extended the study to quantification of infarct sizes [measured by TTC staining] in tau^{-/-} brains 5 days after 90 minutes of transient MCAO. When comparing the infarct size to 24 hours after MCAO, we did not see an increase in mean infarct volumes. The quantification of infarct areas has been included into the revised supplementary figure S4. We also provide now long-term follow up data on tau^{-/-} and wild-type mice that were subjected to 30 minutes of transient MCAO (see new Fig. 2), which showed – consistent with our original findings – a substantial protection from neurological deficits and brain damage in tau^{-/-} mice compared to tau^{+/+} animals. Furthermore, tau^{-/-} recovered faster from neurological and motor deficits, as compared to wild-type animals.

8. Are there any previous studies that investigated the cerebral vascular anatomy in tau^{-/-} mice. If not, the authors are encouraged to present basal and dorsal aspects of the brain vasculature of WT and tau^{-/-} mice as a supplementary figure.

We are not aware of studies into the vascular anatomy in tau^{-/-} mice. As suggested by this reviewer we present basal and dorsal aspects of brains from tau^{-/-} and tau^{+/+} mice perfused with 50% Indian ink/5% gelatin, showing no differences in vascular anatomy. This method has been previously used to visualize the vascular anatomy in rodent brains [Xue et al., *PLOS ONE* 2014]. This data is now presented in the new supplementary figure S1 and referred to in the main text by stating: “There were also no overt differences in the vascular anatomy of the brain between tau^{+/+} and tau^{-/-} mice (fig. S1).” (page 3, lines 16/17).

Reviewer #2

This is a comprehensive assessment of the effects of tau knockout on brain damage from transient middle cerebral artery occlusion (tMCAO) in mice and an assessment of the role of Ras-inhibiting SynGAP1 in the protective effects of tau deletion. Most of the studies appear to be well powered, with robust effects and consistency among the data sets of similar endpoints as well as potential underlying mechanism(s). There are a few issues that the authors need to address which are categorized as major issue and minor issues below.

Major Issues

1. Statistical analysis: The use of student t tests for analysis of complex experimental design is inappropriate. For the described design and results reported, the student t test is an inappropriate statistical approach with which to evaluate group differences for the current series of experiments. Specifically, in the methods section, the authors describe designs involving the assessment of two genotypes (tau^{+/+} or tau^{-/-}), two surgical conditions (Sham or MCAO) and/or two AAV vector conditions (SG1-shR or ctr-shR) on dependent variables of interest, in several instances being assessed at multiple time points following when experimental manipulations were induced. For example, in Figure 1E, the experimental design these data represent is that of a 2 way ANOVA with repeated measures (5 in total). In order to conduct assessments for group differences using t-tests as the author describes, they would have to have conducted 30 individual t-tests to compare each of the groups from each other at each time point. The issue with this is that the authors make no mention of any false discovery rate correction for the increased risk of making a Type I error (incorrectly rejecting the null hypothesis using sample data when there is not a difference in the population) that comes with multiple comparisons. Doing this number of t-test two-group comparisons for a given experiment would yield and 80% that at least one of them is a false positive (for example Fig 1E, but there are numerous other instances of this). The omnibus 2 way ANOVA with follow up contrasts based on the significance of main effects/interactions of these factors would protect against this issue and is the more appropriate approach here. The only instance in Figure 1 where uncorrected t-tests would be appropriate is in Figure 1B/I. This issue of the inappropriateness of the statistical test being applied is present throughout the results section/figures and as such, makes the interpretation of any of the data as they are presented currently a significant challenge and a major concern.

We do appreciate these detailed comments by the reviewer on statistical analysis. As suggested, we have reanalyzed all data using appropriate tests. Details on which tests were used are now provided in each figure legend. Please note that after re-analysis all differences initially reported remained significant, although some *P*-values changed.

2. Tau phosphorylation state: It is not clear from the study description if the knockout of tau protects from stroke damage by removing total tau or prevention of the reported hyperphosphorylation of tau after stroke. This is a critical issue that needs to be addressed by the authors.

We agree with the reviewer that tau hyperphosphorylation may play a role in stroke, particularly in long-term deficits. In acute brain damage following ischemic stroke, however, our study would rather support a role of tau independent of hyperphosphorylation. The crucial finding of the present study supporting this view is that susceptibility of tau^{-/-} mice to

MCAO-induced brain damage and functional deficits was reinstated by reducing SynGAP1 and hence activation of ERK signaling. This brain damage in tau^{-/-} mice with SynGAP1 knockdown is in the absence of tau, and hence tau cannot be phosphorylated. In other words, if tau hyperphosphorylation would be required to mediate the acute brain damage after transient MCAO, then reinstating ERK signaling in tau^{-/-} would have not resulted in increased infarct sizes compared with naive tau^{-/-} mice. Therefore, although this does not entirely exclude that tau hyperphosphorylation contributes to deficits after MCAO to some degree (or later on), ERK signaling appears to play a major role in mediating acute brain damage after transient MCAO. These findings are also in line with previous reports on the role of tau in excitotoxicity; (1) Others and we have previously shown that tau depletion reduces susceptibility to excitotoxicity [Roberson et al., *Science* 2007; Ittner et al., *Cell* 2010; Gheyara et al., *Ann Neurol* 2014]. This is consistent with the protection from excitotoxic brain damage in stroke in tau^{-/-} mice in the present study. (2) We have previously shown that tau is an important part of the post-synaptic signaling complex that mediates excitotoxicity [Ittner et al., *Cell* 2010]. Taken together, our data suggests a mechanistic role of tau via modulation of excitotoxic signaling (including ERK) independent of tau hyperphosphorylation.

Nevertheless, we probed both brain samples from PTZ-injected wild-type mice and wild-type brain sections after transient MCAO with antibodies specific for tau phosphorylated at distinct sites, to put our findings in the context of prior literature on tau phosphorylation after stroke. Using the paradigm of a transient increase in ERK phosphorylation in wild-type mice injected with PTZ (see original Figure 3a), we found some increased phosphorylation of tau at Serine (S) 214, S396/S404 (PHF1) and S422 10 minutes after PTZ administration, that continued to increase 30 minutes after the injection. This suggests that ERK activation precedes tau phosphorylation in this model. Similarly, we found increased tau phosphorylation in the brains of mice 3 hours after transient MCAO. This is in line with previous literature that reported increased tau phosphorylation after stroke in rodent models [REFs]. As expected no tau phosphorylation was detected in brains of tau^{-/-} mice. This data is presented in the new supplementary figure S9 and discussed in the main text by stating: “A number of studies reported changes in tau phosphorylation after ischemia, focusing on long-term effects of pathologically phosphorylated tau (AD-like mechanisms) on memory¹⁴⁻¹⁹. Consistent with these prior studies, we showed increased tau phosphorylation upon PTZ-induced excitotoxicity and MCAO in wild-type mice (fig. S9). However, our findings that reducing SynGAP1 levels and reinstating an excitotoxic ERK response in tau^{-/-} mice increased susceptibility to induced seizures and abolished the protection from MCAO-mediated brain damage despite the absence of tau, supports that tau phosphorylation does not contribute significantly to the acute excitotoxic deficits. A mechanistic role of tau phosphorylation in particular in the long-term deficits following stroke remains to be shown.” (page 9, lines 10-17).

Minor Issues:

1. Figure 1B lacks a description (axis) of the units of cerebral blood flow.

Figure 1B has been revised, and an axis for the units of cerebral blood flow has been included.

2. It is not clear how animals were assigned into the various group after tMCAO. For lesion volume and neurological scores, 12 mice per group were used. For other measurements, much smaller group sizes were used (4-5). Despite these small group sizes, the variances were extremely small of nearly all outcome measures. As such, a description of the assignment of animals to these endpoint measures is needed.

As outlined in the ‘Statistics’ section at the end of the Methods, group sizes for the initial MCAO experiments in tau^{-/-} mice were determined by power calculation: “To detect a 40% reduction in infarct size ($\sigma = 0.2$) with a power of 0.95 and $\alpha = 0.05$ we required a sample size of 8 ($N = 7.61$, Cohen method).” The effect size was greater than expected. Therefore, we used these outcomes to perform sample size power calculations for subsequent experiments: “Based on our tau^{-/-} MCAO data, we calculated a pre-study sample size of 4 ($N = 3.21$) to detect a 75% difference in infarct size for the SynGAP1-knockdown MCAO study.” Error bars show SEM for all experiments.

Randomization of mice was done by an independent staff of the unit neither involved in the experimental design and procedures, nor data analysis and reporting. Note that the mice have no overt phenotype that would allow the experimenter to identify knockout animals. Sufficient mice per genotype were assigned to reach the experimental groups as per above power calculations. Similarly, AAV vials were encoded (i.e. “A” and “B”) to blind the experimenters. We have further clarified the randomization in the revised Methods section: “Experimenters were blinded to the randomly assigned genotype or type of AAV injected for all experiments until after analysis was completed. Blinding and sample/animal randomization was done by staff not involved in the study.” (page 9, lines 34)

3. The sentence on page 7, lines 7 to 8, does not make sense to this reviewer.

We wanted to point out that the infarct size in tau^{-/-} mice 24 hours after transient MCAO was of similar size as in wild-type mice shortly after the transient occlusion. This suggests that the damage observed in tau^{-/-} mice is likely due to the initial ischemia [= core], without further progression of the lesion. We have clarified this in the revised manuscript by

rewording the original statement. It reads now: “For comparison, the brain damage in tau^{-/-} mice 24h after MCAO was of a size similar to that reported for wild-type mice directly after MCAO^{25,28}.” (page 7, line 35/36).

4. Page 8, last paragraph, lines 9-10, is an apparent dismissal of previous work on tau hyperphosphorylation following stroke. This sentence is not needed and should be removed.

As requested by the reviewer, the sentence has been removed. By no means, we intended to mitigate findings by others. The role of tau phosphorylation in the context of this study is now discussed as outlined above (response to major comment 2).

Reviewer #3

The study by Ittner and colleagues demonstrated that tau ablation protected mice from damages caused by MCAO, prevented upregulation of ERK pathways. They also provided evidence that the protection could be mediated by the enhanced SynGAP-PSD95 interaction, upregulated in TauKO mice. The protective effects are striking, and the involvement of SynGAP provided novel mechanistic insight. The overall data quality is excellent. The following concerns need to be addressed:

1. Only male mice are used. Is this phenomena only applicable to males? It would be nice to know.

There is a well-documented sexual dimorphism in rodent stroke models. Female mice (and rats) have significantly smaller infarcts following MCAO [Simpkins JW et al., *Cellular and Molecular Life Sciences*, 2005]. This protection is thought to be mediated by estrogen [McCullough LD and Hurn PD, *Trends in Endocrinology and Metabolism*, 2003], and is reduced as female animals age [Liu F and McCullough LD, *Journal of Biomedicine and Biotechnology*, 2011]. To obtain reproducible large tissue damage after MCAO, the majority of pre-clinical studies use male rodents. In order to determine the protective effects of tau reduction on brain damage after transient MCAO, we also used only male mice.

2. Inhibition of SynGAP1 makes the tauKO mice vulnerable to MCAO. One concern is that SynGAP1 inhibition could be detrimental, thus the elevated toxicity could contribute to MCAO damage in tauKO mice via a parallel pathway. A complementary experiment could be more informative. Does overexpressing synGAP1 in tau+/+ mice render them protective against MCAO?

We agree with this reviewer that the reciprocal experiment with overexpression of SynGAP1 in wild-type mice to mediate protection from MCAO-induced brain damage would be interesting. Using AAV-mediated expression would have allowed us to perform MCAO experiments in wild-type mice with overexpression of SynGAP1 within the timeframe of a paper revision. Unfortunately, the cDNA encoding SynGAP1 is 4031bp, which is beyond the packaging capacity of neuronal AAV expression vectors. Nevertheless, we attempted to package and express SynGAP1 by using a truncated synapsin promoter and a minimal poly-adenylation sequence. All constructs were sequenced. Still, SynGAP1 could not be over-expressed in primary neurons and mice via AAV. Hence, the suggested experiments could not be performed. Note that the cDNA encoding SynGAP1 resulted in expression of the protein when transiently expressed in 293T cells using a standard pcDNA expression vector, suggesting that transcription/translation from the used cDNA worked in principle.

The only feasible way forward would be the generation of a novel SynGAP1 transgenic mouse line by pronuclear injection. However, this would take well more than one year to generate and analyze. We hope that the reviewer agrees that this is beyond the scope of this revision and should be subject to a follow-up study with a new mouse line.

Minor comments:

1. The legends include non-necessary information (i.e discussion of the results etc.). Consider revise to be more concise.

As requested by the reviewer, we have shortened the legends to the main figures, retaining sufficient information to guide the reader through the presented data. We feel that any further truncation would complicate the readability.

2. The genes in Fig. 2C are not readable. Consider revising.

We do recognize that the gene names are hard to read and require zooming in. We have highlighted in the figure legend that the genes presented in the heat map (now Fig. 3c) are listed in Table S2. Since the journal is digital only, we would prefer to leave the gene names next to the heat map, allowing readers to zoom in and directly identify each gene. We leave it to the editor and reviewer to decide if the gene names should be removed from the main figure.

Reviewer #4 [Additional feedback]

The (stroke) data are extremely limited because of the lack of an extended time course and rudimentary outcomes (NSS only for behavior). It's unclear why both groups had such limited outcomes in terms of time course in all outcomes and behavior techniques. Long term outcomes and functional tests are the essential requirements for stroke research nowadays. And with the NSS data, it's surprising to see the score of 2 extending to 120h post-MCAO for the $-/-$ mice. As this measures limb extension, this score is typically around zero by 3-5 days after MCAO with such small infarct. The study should show the 120 h timecourse for the $+/+$ mice. In addition, the entire Fig. 4 was done using the $-/-$ mice. The study should include the wildtype tau mice for comparisons.

Extended time course: In the revised manuscript, we now provide extended time course of both $\tau^{-/-}$ and $\tau^{+/+}$ mice after MCAO. In addition to the neurological scoring (NSS) over time, we also subjected the mice to objective motor testing. Specifically, $\tau^{-/-}$ and $\tau^{+/+}$ mice were followed for 2 weeks after 30 minutes MCAO, until $\tau^{+/+}$ mice returned to a neurological score of zero. Consistent with our previous findings that $\tau^{-/-}$ are protected from severe neurological deficits, NSS was lower in $\tau^{-/-}$ mice compared to $\tau^{+/+}$ animals early on after MCAO, and improved significantly faster to no overt neurological deficits 4 days after the procedure, 2 days earlier than in $\tau^{+/+}$ mice. Similarly, $\tau^{-/-}$ mice performed significantly better on the accelerating Rota-Rod 2 days after MCAO compared to $\tau^{+/+}$ animals. Both, $\tau^{-/-}$ and $\tau^{+/+}$ mice performed similar to prior to the surgeries after 6 days. Body weight dropped significantly more after MCAO and recovered much slower in $\tau^{+/+}$ mice compared to $\tau^{-/-}$ animals. This new data is now presented in the new Figure 2 and discussed in the main text by stating: “The substantial brain damage and profound functional deficits in $\tau^{+/+}$ mice after 90 minutes of MCAO did not allow following up $\tau^{+/+}$ mice for longer than 24h. Therefore, we subjected an additional cohort of $\tau^{+/+}$ and $\tau^{-/-}$ mice to 30 minutes of MCAO allowing longer-term follow up experiments (Fig. 2a). Following this milder transient MCAO, $\tau^{-/-}$ mice displayed significantly less severe neurological deficits compared to $\tau^{+/+}$ mice receiving this treatment (Fig. 2b). Both improved their functional deficits over the following days, with $\tau^{-/-}$ displaying no neurological deficits on day 4 after MCAO, while $\tau^{+/+}$ mice required 6 days to fully recover. Similarly, $\tau^{-/-}$ mice lost significantly less body weight within 2 days after MCAO, and recovered the lost weight faster than $\tau^{+/+}$ animals (Fig. 2c). Notably, $\tau^{+/+}$ did not fully recover the lost body weight within 2 weeks after 30 minutes of transient MCAO. Both, $\tau^{+/+}$ and $\tau^{-/-}$ mice showed significantly decreased performance on the accelerating Rota-Rod 2 days after the MCAO procedure, although $\tau^{-/-}$ mice performed significantly better than $\tau^{+/+}$ mice (Fig. 2d). These deficits recovered fully 4 and 6 days after MCAO in $\tau^{-/-}$ and $\tau^{+/+}$ mice, respectively. As expected, the brain damage was substantially less in $\tau^{-/-}$ than $\tau^{+/+}$ mice 14 days after 30 minutes of transient MCAO (Fig. 2e,f). Hence, reduced neurological and functional deficits were less profound and recovery was faster in $\tau^{-/-}$ than $\tau^{+/+}$ mice after 30 minutes of transient MCAO, and was associated with markedly smaller infarcts.” (page 4, lines 8-21). We note that the 90 minutes MCAO used in the previous experiments resulted in seriously compromised wellbeing of wild-type mice within 24 hours after the surgery, which need to be terminated to meet ethical requirements. Therefore, we reduced the duration of the MCAO to 30 minutes, which guaranteed long-term survival of wild-type mice without ethical concerns of compromised wellbeing.

Neurological scoring: We do appreciate that there are differences in NSS assessment across studies. In the figure to the editor below, we provide pictures of mice from our study and how they correspond to the neurological scores of ‘1’ to ‘4’ used in our study. Note that any asymmetry (visibility of the left front paws from above; see image on the far left) was scored as ‘2’. This was done irrespectively of the mice showing resistance to lateral push (score of 1) or not. We do realize that an escalation scoring would have reflected these improvements better. We therefore rescored all mice that were followed up for longer than 24 hours to reflect their recovery of motor coordination and strength; accordingly, mice that did not show resistance to lateral push were scored as ‘0’, even if they showed slight asymmetry of the forelimb. The new scoring is now presented in the revised fig. S4, showing the improvement over time to very minimal mean impairments 5 days after 90 minutes of transient MCAO. The main test has been changed accordingly: “ $\tau^{-/-}$ mice observed for up to 5 days after transient MCAO did show progressive improvement of the mild neurological deficits with no increase in infarct volumes (fig. S4), suggesting persisting protection.” (page 4, lines 4-5). We would like to thank the reviewer for the advice on this matter. We leave it to the editor and reviewer whether the below figure should be included in the supplementary material.

Figure to the reviewer | Example of mice with neurological scores (NSS) of ‘1’ to ‘4’.

SynGAP1 knockdown in $\tau^{+/+}$ mice: As requested by this reviewer, we now present data from wild-type mice after AAV-shRNA-mediated knockdown of SynGAP1. Given the substantial deficits and brain damage of wild-type mice after 90 minutes of transient MCAO, knockdown of SynGAP1 (using AAV-mediated shRNA expression) did not result in a significant worsening of outcomes upon reduction of SynGAP1 levels. Accordingly, there was a moderate trend to a reduced latency to develop more severe seizures in wild-type mice with AAV-mediated shRNA expression targeting SynGAP1 (AAV-SG1-shR) compared to controls (AAV-ctr-shR). Also, more AAV-SG1-shR injected wild-type mice progressed to convulsive seizures than those having received AAV-ctr-shR. Similarly, there was a trend to higher levels of ERK phosphorylation 10 minutes after PTZ in AAV-SG1-shR compared to AAV-ctr-shR injected wild-type mice. Neurological deficits 24 hours after transient MCAO were similarly severe in AAV-SG1-shR and AAV-ctr-shR injected wild-type mice. Finally, the infarct size were comparable 24 hours after transient MCAO in in AAV-SG1-shR compared to AAV-ctr-shR injected wild-type mice, likely due to already substantial infarcts in controls. This new data is presented in the new supplementary figure S8 and discussed in the main manuscript by stating: “In $\tau^{+/+}$ mice, there was a moderate trend to a reduced latency to develop more severe seizures with more animals progressing to convulsive seizures when injected with AAV-SG1-shR compared with AAV-ctr-shR-injected controls (fig. S8). Correspondingly, AAV-SG1-shR-injected $\tau^{+/+}$ mice showed a trend towards higher levels of ERK phosphorylation 10 minutes after PTZ compared to AAV-ctr-shR-injected controls (fig. S8).” (page 7, lines 6-10), and “Infarct sizes were comparable 24 hours after transient MCAO in in AAV-SG1-shR-injected $\tau^{+/+}$ mice compared to AAV-ctr-shR-injected controls, and their neurological deficits did not worsen any further, likely due to the already profound deficits of controls (fig. S8).” (page 7, lines 16-19)

Reviewers' comments:

Reviewer #1 (Remarks to the Author):

I wanted to compliment the authors on the quality of the revised manuscript and on the rigor of addressing all of the reviewer's comments. The additional experiments performed for this resubmission make sense and clearly increase the impact of this study. Two minor comments to be addressed are:

Page 10; "Coordinates for cortical injections were: AP: -2mm, RL: 1.2mm, RC: 1mm." RC (I assume it stands for rostral/caudal) should be replaced by DV (dorsal/ventral) for rodents.

Figure S1. It would seem that Tau^{-/-} mice have a non-patent PCOM compared to Tau^{+/+} mice. It could be also a perfusion artifact because dorsal cerebellar arteries are also less presented in the Tau^{-/-} brain. If rudimentary PCOM was not a commonly found feature in Tau^{-/-} mice, the authors should depict a brain that is more representative of the anatomy in Tau^{-/-} mice. From a pathophysiological perspective, obviously it does not make sense. A non-patent PCOM should render Tau^{-/-} mice more susceptible to ischemic brain damage.

Reviewer #3 (Remarks to the Author):

The revised manuscript is much improved. The data presented are interesting and of good quality. There are remaining concerns that need to be addressed:

1) The model in Figure 5j is misleading because it suggests that tau binding to PSD95 prevents the interaction of SynGAP1 with PSD95. This is an interesting possibility, however they don't have any data to support it yet.

2) The authors pointed out that they were unable to express SynGAP1 in mouse brain by virus because of its size. To support their conclusions about the signaling pathway, it is important to overexpress SynGAP1 to rescue excitotoxicity. They could at least transfect cultured neurons with SynGAP1 and show whether NMDAR-mediated ERK phosphorylation is blocked by looking at p-ERK immunocytochemistry (example: Huang, Chotiner and Steward, J Neuro 2007).

Point-to-point response to reviewer comments

We would like to thank the reviewers for their additional comments. We have addressed all of the comments as outlined below, including new experiments where requested. We hope that the reviewers are now able to recommend the manuscript for publication.

Reviewer #1

I wanted to compliment the authors on the quality of the revised manuscript and on the rigor of addressing all of the reviewer's comments. The additional experiments performed for this resubmission make sense and clearly increase the impact of this study. Two minor comments to be addressed are:

Page 10; "Coordinates for cortical injections were: AP: -2mm, RL: 1.2mm, RC: 1mm." RC (I assume it stands for rostral/caudal) should be replaced by DV (dorsal/ventral) for rodents.

This has been changed to DV in the revised manuscript. It reads now: "AP: -2mm, RL: 1.2mm, DV: 1mm." (page 10, line 26)

Figure S1. It would seem that Tau^{-/-} mice have a non-patent PCOM compared to Tau^{+/+} mice. It could be also a perfusion artifact because dorsal cerebellar arteries are also less presented in the Tau^{-/-} brain. If rudimentary PCOM was not a commonly found feature in Tau^{-/-} mice, the authors should depict a brain that is more representative of the anatomy in Tau^{-/-} mice. From a pathophysiological perspective, obviously it does not make sense. A non-patent PCOM should render Tau^{-/-} mice more susceptible to ischemic brain damage.

As suggested by the reviewer, as more representative Tau^{-/-} brain is now presented in the Supplementary Figure 1. We did not observe obvious differences in the PCOM in Tau^{+/+} and Tau^{-/-} mice.

Reviewer #3

The revised manuscript is much improved. The data presented are interesting and of good quality. There are remaining concerns that need to be addressed:

1) The model in Figure 5j is misleading because it suggests that tau binding to PSD95 prevents the interaction of SynGAP1 with PSD95. This is an interesting possibility, however they don't have any data to support it yet.

We agree with the reviewer that our data only provided evidence for increased SynGAP1 in complex with PSD-95 in the absence of tau, while our previous study and others showed that tau and PSD-95 interact (Ittner L et al., Cell 2010; Mondragon-Rodriguez S et al., JBC 2012). To provide direct evidence for tau preventing SynGAP1 from binding to PSD-95, we expressed combination of PSD-95, tau and SynGAP1 in cells and co-immunoprecipitated the complexes. We found that markedly less SynGAP1 interacted with PSD-95 when tau was co-expressed. This suggests that indeed, tau binding to PSD-95 reduces additional binding of SynGAP1. This new data is now provided in the revised Figure 4p and referred to in the text by stating: "Finally, SynGAP1 co-immunoprecipitate with PSD-95 in the absence, but not in the presence of co-expressed tau from transiently transfected cells (Fig. 4p), suggesting negative regulation of PSD-95/SynGAP1 complexes by tau." (page 6, lines 21-23) Based on this new data, we have decided to not alter the model presented in Figure 5j. However, we leave it to the editor/reviewer if they prefer to remove the model.

2) The authors pointed out that they were unable to express SynGAP1 in mouse brain by virus because of its size. To support their conclusions about the signaling pathway, it is important to overexpress SynGAP1 to rescue excitotoxicity. They could at least transfect cultured neurons with SynGAP1 and show whether NMDAR-mediated ERK phosphorylation is blocked by looking at p-ERK immunocytochemistry (example: Huang, Chotiner and Steward, J Neuro 2007).

As suggested by the reviewer, we have expressed SynGAP1 in primary neurons and determined the effects on NMDA-induced ERK phosphorylation. Expression of SynGAP1 blocked the phosphorylation of ERK in response to NMDA. This new data is now presented in the new Supplementary Figure 10 and referred to in the text by stating: "Conversely, SynGAP1 over-expression in primary neurons mitigated NMDA-induced ERK phosphorylation (Supplementary Fig. 10)." (page 7, lines 15/16)

REVIEWERS' COMMENTS:

Reviewer #3 (Remarks to the Author):

The authors have adequately addressed this reviewer's comments.

Point-to-point response to reviewer comments

There were no outstanding reviewer comments to be addressed.